# ZEMBA v1.0: an energy and moisture balance climate model to investigate Quaternary climate

**Daniel F. J. Gunning**[1], **Kerim H. Nisancioglu**[1], **Emilie Capron**[2], **and Roderik S. W. van de Wal**[3,4]

[1]Department of Earth Science, University of Bergen and Bjerknes Centre for Climate Research, Bergen, Norway
[2]Université Grenoble Alpes, CNRS, INRAE, IRD, Grenoble INP, IGE, 38000 Grenoble, France
[3]Institute for Marine and Atmospheric research Utrecht, Utrecht University, Utrecht, the Netherlands
[4]Department of Physical Geography, Faculty of Geosciences, Utrecht University, Utrecht, the Netherlands TS1

**Correspondence:** Daniel F. J. Gunning (daniel.gunning@uib.no)

**Abstract.** The Zonally Averaged Energy and Moisture BAlance (ZEMBA) climate model is introduced as a simple and computationally efficient tool for studies of the glacial–interglacial cycles of the Quaternary. The model is based on an energy balance model comprising an atmospheric layer, a land component and a two-dimensional ocean transport model with sea ice. In addition, ZEMBA replaces temperature with moist static energy for calculations of diffusive heat transport in the atmospheric layer and includes a hydrological cycle for simulating precipitation and snowfall. Prior to coupling with an ice sheet model, we present and evaluate equilibrium simulations of the model for the pre-industrial period and the Last Glacial Maximum, using prescribed land ice fractions and elevation. In addition, we test the sensitivity of ZEMBA to a doubling of the atmospheric $CO_2$ concentration and a 2 % increase in solar radiation at the top of the atmosphere. Compared to a global climate model (the Norwegian Earth System Model version 2, NorESM2) and reanalysis data (ERA5), ZEMBA reproduces the zonally averaged climate of the pre-industrial period with reasonable accuracy, capturing features such as surface temperature, precipitation, radiative fluxes, snow cover, sea ice cover and meridional heat transport. The response of ZEMBA to increasing $CO_2$ concentrations is qualitatively similar to the observational record and climate models of higher complexity, including polar amplification over the Northern Hemisphere and during the winter months. The globally averaged rise in surface air temperature for a doubling in $CO_2$ is 3.6 °C. Finally, ZEMBA shows success in emulating changes in surface temperature and precipitation during the Last Glacial Maximum when compared to reconstructions and global climate models.

## 1 Introduction

Since the beginning of the Quaternary period (2.58 Ma to present), the Earth's climate has repeatedly switched between cold "glacial" periods and warmer "interglacial" periods, which are collectively known as glacial–interglacial cycles. Glacial periods are characterized by the presence of large ice sheets covering North America and Fennoscandia, while interglacials refer to times when ice sheets are restricted to Greenland and Antarctica, such as the current Holocene period (11.7 ka to present). The glacial–interglacial cycles are widely documented in natural climate archives. For example, the oxygen isotope ratio ($\delta^{18}O$) recorded in the shells of micro-organisms that accumulated on the ocean floor – hereafter referred to as benthic $\delta^{18}O$ – serves as a valuable proxy for global ice volume and deep-ocean temperatures spanning millions of years into the past (Lisiecki and Raymo, 2005; Elderfield et al., 2012). Additionally, Antarctic ice cores contain valuable information over the past 800 kyr such as atmospheric $CO_2$ concentrations (Bereiter et al., 2015) and surface temperature changes (Jouzel et al., 2007; Kawamura et al., 2017).

Despite extensive research, a comprehensive understanding of what caused the glacial–interglacial cycles has remained elusive, although the importance of changes in the Earth's orbital parameters has been widely acknowledged.

Changes in these orbital parameters are thought to determine when the Earth switches between glacial and interglacial climates by redistributing the incoming solar radiation (insolation) the Earth receives across latitudes and seasons. These variations in the orbital parameters encompass changes in the shape of the Earth's orbit (eccentricity) on 100 and 413 kyr cycles, changes in the tilt (obliquity) of the Earth's rotational axis on 41 kyr cycles, and changes in the time of the year when the Earth is closest to the Sun (climatic precession) on 19 and 23 kyr cycles. The most favoured of the orbital hypotheses comes from Milutin Milanković, who proposed that glaciation occurs during times of reduced insolation at the high northern latitudes during the summer months, when obliquity is low and the Northern Hemisphere (NH) summers coincide with the Earth's furthest distance from the Sun (and vice versa for deglaciation). Hays et al. (1976) were the first to discover that benthic $\delta^{18}$O records contained cycles with periods of 23, 41 and 100 kyr, in correspondence to the Earth's orbital cycles. Subsequent research has extended the benthic $\delta^{18}$O record further back in time (Pisias and Moore, 1981; Ruddiman et al., 1986) and led to compiled records from across the globe (Ahn et al., 2017; Lisiecki and Raymo, 2005), which together with other proxy records (Jouzel et al., 2007; Kawamura et al., 2017) provides compelling evidence for an orbital control on climate change. Nevertheless, despite a distinct orbital "rhythm" to glacial–interglacial cycles, important characteristics of the benthic $\delta^{18}$O record cannot be easily explained by Milanković theory, including a shift in the dominant periodicity observed around 1 Ma.

The Mid-Pleistocene Transition (MPT) from 1.25 to 0.7 Ma represents a transition in the dominant periodicity of glacial–interglacial cycles from 41 kyr in the Early Pleistocene to $\sim 100$ kyr in the Late Pleistocene, occurring without significant variations in the orbital cycles (Clark et al., 2006). Hypotheses for the MPT are abundant in the literature (Berends et al., 2021). Prominent theories include the removal of a subglacial regolith beneath the NH ice sheets (Clark and Pollard, 1998; Clark et al., 2006), a gradual decline in atmospheric $CO_2$ concentrations (Raymo et al., 1988; Berger et al., 1999), the merging of the Laurentide and Cordilleran ice sheets over North America (Bintanja and Van De Wal, 2008), phase locking between Antarctica and the NH ice sheets (Raymo et al., 2006), or some combination of these mechanisms (Chalk et al., 2017; Willeit et al., 2019). Even the $\sim 100$ kyr cycles in glacial ice volume following the MPT were not an expected outcome of Milanković theory, as they correspond to changes in the Earth's eccentricity, which has a negligible direct influence on summer insolation (Imbrie et al., 1993). Instead, these $\sim 100$ kyr cycles have been explained as the skipping of one (80 kyr) or two (120 kyr) obliquity cycles (averaging to 100 kyr), leading to longer glacial cycles in the Late Pleistocene (Huybers and Wunsch, 2005), where the precise timing of deglaciation is set by the combined forcing of precession and obliquity (Huybers, 2011; Parrenin and Paillard, 2012). Independent mod-

elling studies have also highlighted the importance of precession, and its modulation by eccentricity, for generating the $\sim 100$ kyr cycles of the Late Pleistocene (Abe-Ouchi et al., 2013; Ganopolski and Calov, 2011). Interestingly, models that have simulated the MPT and/or the $\sim 100$ kyr cycles of the Late Pleistocene still struggle to reproduce the dominant 41 kyr cycles seen in the Early Pleistocene (Berger et al., 1999; Willeit et al., 2019; Watanabe et al., 2023).

While modelling experiments of the Early Pleistocene capture 41 kyr cycles in ice volume, in correspondence to the Earth's obliquity, they generate stronger 19 and 23 kyr precession cycles (Berger et al., 1999; Willeit et al., 2019; Watanabe et al., 2023) than observed in the benthic $\delta^{18}$O record (Lisiecki and Raymo, 2005). This is unsurprising given that climatic precession controls the intensity of summer insolation. Indeed, for various metrics of summer insolation variability, including mid-month insolation (i.e. 21 June), monthly mean insolation or the caloric summer half year, precession has a strong influence at latitudes where NH ice sheets grow and melt. It is worth noting that precession cycles are detectable prior to the MPT (Lisiecki and Raymo, 2007; Liautaud et al., 2020), but they become more pronounced across the Quaternary, with much stronger signals observed in the Late Pleistocene compared to the Early Pleistocene (Raymo and Nisancioglu, 2003). Theories that account for weaker 19 and 23 kyr precession cycles during the Early Pleistocene include a counterbalancing between summer insolation intensity and summer duration (Huybers, 2006; Huybers and Tziperman, 2008), the cancellation of out-of-phase precession cycles between the NH ice sheets and Antarctica (Raymo et al., 2006; Morée et al., 2021), or the influence obliquity has on the poleward flux of moisture and accumulation rates on ice sheets (Raymo and Nisancioglu, 2003; Nisancioglu, 2004).

Models of varying levels of complexity have been employed to address these questions relating to glacial–interglacial cycles. Global climate models (GCMs) are too computationally expensive for simulations on these timescales, so studies have instead relied on conceptual models (Paillard, 1998; Parrenin and Paillard, 2003; Legrain et al., 2023), energy balance models (Pollard, 1978; Huybers and Tziperman, 2008), Earth system models of intermediate complexity (Ganopolski and Calov, 2011; Willeit et al., 2019), climate parameterization based on discrete GCM "snapshots" (Abe-Ouchi et al., 2013) or benthic $\delta^{18}$O records (Bintanja and Van De Wal, 2008). Among these, zonally averaged energy balance models (EBMs) provide computationally efficient tools for studying the response of the Earth's climate to changes in the Earth's orbital parameters (Suarez and Held, 1979; Pollard, 1978; Huybers and Tziperman, 2008; Stap et al., 2014). EBMs calculate the distribution of surface temperature with latitude by considering the conservation of energy on a sphere subject to heating by solar insolation, cooling by terrestrial radiation and the diffusive redistribution of heat from the Equator to the poles. Since the

original works of Budyko (1969) and Sellers (1969), EBMs have long been used to study the Earth's climate sensitivity and have shown success at simulating both present-day and glacial climate states (North, 1975; Peng et al., 1987; Harvey, 1988; Jentsch, 1991; Bintanja and Oerlemans, 1996; Bintanja, 1997; Stap et al., 2014). Moreover, the simplicity of EBMs enables the isolation and identification of important processes and feedbacks (Huybers and Tziperman, 2008; Stap et al., 2014). Consequently, EBMs provide valuable tools for exploring the glacial–interglacial cycles of the Quaternary and guiding further investigations with more realistic models.

In this study, we introduce the Zonally Averaged Energy and Moisture BAlance (ZEMBA) climate model to study the response of the Earth's climate to changes in the orbital parameters. The model is designed to place a particular emphasis on physical processes that may influence the relative contributions of obliquity and precession to climate variability during periods such as the Early Pleistocene. The model spans both hemispheres and is forced by the full seasonal cycle in insolation. Unlike previous EBMs used for studies of glacial–interglacial cycles (Pollard, 1978; Huybers and Tziperman, 2008; Stap et al., 2014), ZEMBA includes a hydrological cycle to simulate precipitation and snowfall. Before using ZEMBA in experiments relating to glacial–interglacial cycles, it is important to ensure the model can simulate the present-day (or pre-industrial) climate with reasonable accuracy and to constrain the sensitivity of the model. Consequently, the following sections document the model and its climate sensitivity. Firstly, we describe a "control" simulation of ZEMBA for the pre-industrial period and compare the broad-scale features with a GCM and atmospheric reanalysis (Sect. 3.1). Then, we test the sensitivity of the model for a doubling of the atmospheric $CO_2$ concentration and a 2 % increase in solar insolation (Sect. 3.2). Finally, we evaluate its performance for a simulation of the Last Glacial Maximum (LGM) (Sect. 3.3). Prior to that, we provide a detailed description of the atmospheric (Sect. 2.1), land (Sect. 2.2) and ocean (Sect. 2.3) components of ZEMBA.

## 2 Model description

ZEMBA is primarily based on the EBM from Bintanja (1997), which comprises a single atmospheric layer overlying a surface divided into land and ocean. While utilizing the same shortwave and longwave radiation scheme as in Bintanja (1997), in addition to the same ocean transport model, ZEMBA includes a hydrological cycle to estimate precipitation and snowfall rates. Moreover, instead of parameterizing the surface albedo over land as a function of surface temperature, ZEMBA directly estimates snow coverage through the competition between snow accumulation (from the hydrological cycle) and ablation (from the surface energy balance).

Following recent studies (Hwang and Frierson, 2010; Rose et al., 2014; Roe et al., 2015; Feldl and Merlis, 2021), atmospheric heat transport is now proportional to meridional gradients in near-surface moist static energy – instead of classic (or "dry") EBMs that diffuse heat along temperature gradients. Moreover, a Hadley cell parameterization is included to produce an equatorward flux of moisture in the tropics (Siler et al., 2018). The radiative and turbulent heat fluxes are calculated separately over land and ocean, but atmospheric temperatures and humidities are set to the zonal average of land and ocean at the end of each model time step. The subsequent sections will describe in more detail the individual atmospheric, land and ocean components of ZEMBA.

### 2.1 Atmosphere

A vertically averaged atmospheric layer simulates the evolution of near-surface air temperature ($T_a$) through time. It evolves as a function of the radiative fluxes exchanged at the top and bottom of the atmospheric layer, the turbulent exchange of heat with the surface layer, the latent heat released during precipitation and snowfall, and the divergence of atmospheric heat transport. The temporal evolution of $T_a$ is described as follows:

$$C_a \, H_a \, \rho_a \, \frac{\delta T_{a(i)}}{\delta t} = S_{a(i)} + I_{a(i)} + K_{(i)} + L_v P_{r(i)} + L_f P_{s(i)}$$
$$+ \frac{1}{2\pi R_e^2 \cos(\theta)} \frac{\delta F_T}{\delta \theta}, \tag{1}$$

where $C_a$, $H_a$ and $\rho_a$ are the specific heat, height and density of the atmospheric layer, respectively; $i$ is the index representing either land or ocean grid cells; $S_a$ is the absorbed shortwave radiation; $I_a$ is the absorbed longwave radiation; $K$ is the exchange of sensible heat with the surface; $L_v P_r$ is the latent heat released during precipitation, where $L_v$ is the latent heat of vaporization and $P_r$ is precipitation; and $L_f P_s$ is the latent heat released during snowfall, where $L_f$ is the latent heat of fusion and $P_s$ is snowfall. Finally, the last term on the right side of the equation represents the horizontal diffusion of temperature, where $R_e$ is the Earth's radius, $\theta$ is the latitude and $F_T$ is the northward flux of dry static energy in the atmospheric layer. All terms in Eq. (1) are in $\mathrm{W\,m^{-2}}$.

#### 2.1.1 Radiative fluxes

The model is forced by diurnally averaged solar insolation at the top of the atmosphere ($S_{\mathrm{TOA}}^{\downarrow}$) using the orbital parameter solution from Laskar et al. (2004). The seasonal cycle is driven exclusively by changes in insolation. The absence of a seasonal insolation cycle results in a markedly colder climate. As noted by Bintanja (1997), employing an annual-mean version of their EBM results in insolation no longer being concentrated in the summer months, when lower zenith angles and reduced snow cover enhance the absorption of shortwave radiation. The amount of solar radiation that is reflected at the top of the atmosphere ($S_{\mathrm{TOA}}^{\uparrow}$), transmitted to the

surface ($S^{\downarrow}_{\mathrm{BOA}}$) and reflected at the surface ($S^{\uparrow}_{\mathrm{BOA}}$) is calculated using the parameterization from Bintanja (1996). The shortwave parameterization takes into account several atmospheric properties including surface air temperature, surface albedo, solar zenith angles, cloud optical depth and surface height. The shortwave fluxes are computed for both clear-sky and overcast conditions, with the total radiative flux for a given grid cell determined as the weighted average using prescribed cloud cover fractions. In this study, cloud cover fractions over land and ocean are taken from pre-industrial simulations of the Norwegian Earth System Model version 2 (NorESM2; Seland et al., 2020) (Fig. 1a). Daytime-mean solar zenith angles are calculated with equations provided by Balmes and Fu (2020). Of particular importance to the shortwave radiative fluxes is the cloud optical depth parameter ($\tau$). Following Bintanja (1996, 1997) and Stap et al. (2014), $\tau$ is kept fixed to a globally and seasonally invariant value. The amount of shortwave radiation that is absorbed by the atmosphere ($S_{\mathrm{a}}$) and the surface ($S_{(i)}$) – either land or ocean – is described below:

$$S_{(i)} = S^{\downarrow}_{\mathrm{BOA}(i)} - S^{\uparrow}_{\mathrm{BOA}(i)}, \tag{2}$$

$$S_{\mathrm{a}(i)} = (S^{\downarrow}_{\mathrm{TOA}(i)} - S^{\uparrow}_{\mathrm{TOA}(i)}) - S_{(i)}. \tag{3}$$

The outgoing longwave radiative fluxes at the TOA ($I^{\uparrow}_{\mathrm{TOA}}$) and the surface ($I^{\uparrow}_{\mathrm{BOA}}$), together with the incoming longwave flux at the surface ($I^{\downarrow}_{\mathrm{BOA}}$), are also calculated using a radiation parameterization from Bintanja (1996). The longwave parameterization is made a function of surface temperature, surface air temperature, cloud emissivity, surface elevation and atmospheric $CO_2$ concentrations. As with the shortwave parameterization, the longwave radiative fluxes are calculated separately for clear-sky and overcast conditions. Alterations to the longwave parameterization from Stap et al. (2014), to both increase the climate sensitivity per $CO_2$ doubling and account for the effects of non-$CO_2$ greenhouse gases, are maintained. The absorbed longwave radiation fluxes at the surface ($I_{(i)}$) and by the atmosphere ($I_{\mathrm{a}}$) are shown below:

$$I_{(i)} = I^{\downarrow}_{\mathrm{BOA}(i)} - I^{\uparrow}_{\mathrm{BOA}(i)}, \tag{4}$$

$$I_{\mathrm{a}(i)} = I^{\uparrow}_{\mathrm{BOA}(i)} - (I^{\downarrow}_{\mathrm{BOA}(i)} + I^{\uparrow}_{\mathrm{TOA}(i)}). \tag{5}$$

### 2.1.2 Turbulent heat fluxes

The aerodynamic bulk relationships are employed to compute the fluxes of sensible heat ($K$) and evaporation ($E$) across the atmosphere–ocean and atmosphere–land interfaces:

$$K_{(i)} = \rho_{\mathrm{a}} \, c_{\mathrm{a}} \, \kappa_{(i)} \left( T_{(i)} - T_{\mathrm{a}(i)} \right), \tag{6}$$

$$E_{(i)} = \rho_{\mathrm{a}} \, W_{(i)} \, \kappa_{(i)} \left( Q_{\mathrm{sat}}(T_{(i)}) - Q_{\mathrm{a}(i)} \right), \tag{7}$$

where $\kappa$ is the turbulent exchange coefficient; $T_{(i)}$ is the temperature of the surface – either land or ocean; $W$ denotes the surface water availability; $Q_{\mathrm{sat}}(T_{(i)})$ is the saturation specific humidity of the surface (as determined by the Clausius–Clapeyron relation); and $Q_{\mathrm{a}}$ is the specific humidity of the overlying atmospheric layer. The influence of wind speed and surface roughness on the turbulent heat exchange is not incorporated into $\kappa$, which remains constant across latitudes and seasons. Following Bintanja (1997), $W$ is set to 0.7 and 1.0 over land and ocean, respectively, to reflect reduced water availability over land. The latent heat flux associated with evaporation is simply $L_{\mathrm{v}} E$.

### 2.1.3 Hydrological cycle

An atmospheric moisture budget is introduced to parameterize the hydrological cycle and simulate precipitation and snowfall (Fanning and Weaver, 1996; Robinson et al., 2010; Ritz et al., 2011):

$$\rho_{\mathrm{a}} H_{\mathrm{a}} \frac{\delta Q_{\mathrm{a}(i)}}{\delta t} = (E_{(i)} - P_{\mathrm{r}(i)}) + \frac{1}{2\pi R_{\mathrm{e}}^2 \cos(\theta)} \frac{\delta F_Q}{L_{\mathrm{v}} \delta \theta}, \tag{8}$$

where the first term on the right side of the equation represents the sources (evaporation) and sinks (precipitation) and the second term represents the horizontal transport of water vapour, in which $F_Q$ is the latent heat associated with the northward moisture flux. Precipitation occurs once the relative humidity, $r$, exceeds a maximum threshold, $r_{\mathrm{max}}$:

$$P_{\mathrm{r}(i)} = \begin{cases} \frac{\rho_{\mathrm{a}} \, H_{\mathrm{a}}}{3 \cdot \Delta t} \left( Q_{\mathrm{a}(i)} - r_{\mathrm{max}} \cdot Q_{\mathrm{sat}}(T_{\mathrm{a}(i)}) \right), & r > r_{\mathrm{max}}, \\ 0, & \text{otherwise,} \end{cases} \tag{9}$$

where $Q_{\mathrm{sat}}(T_{\mathrm{a}})$ is the saturation specific humidity of the atmospheric layer and $\Delta t$ is the seconds in 1 d. When the relative humidity exceeds $r_{\mathrm{max}}$, it leads to precipitation. The turnover time for excess humidity in the atmosphere is set to 3 d. Shorter turnover times produce very large and sporadic contributions of latent heat to the atmospheric column. The amount of snowfall is determined using

$$P_{\mathrm{s}(i)} = P_{\mathrm{r}(i)} \cdot f_{\mathrm{sf}}, \tag{10}$$

where $f_{\mathrm{sf}}$ is the fraction of precipitation that falls as snow, which is parameterized as a function of surface air temperature (Harvey, 1988): TS2

$$f_{\mathrm{sf}(i)} = \begin{cases} 1, & T'_{\mathrm{a}(i)} < 260\,\mathrm{K}, \\ 0.005(280 - T'_{\mathrm{a}(i)}), & 260\,\mathrm{K} \leq T'_{\mathrm{a}(i)} \leq 280\,\mathrm{K}, \\ 0, & T'_{\mathrm{a}(i)} > 280\,\mathrm{K}, \end{cases} \tag{11}$$

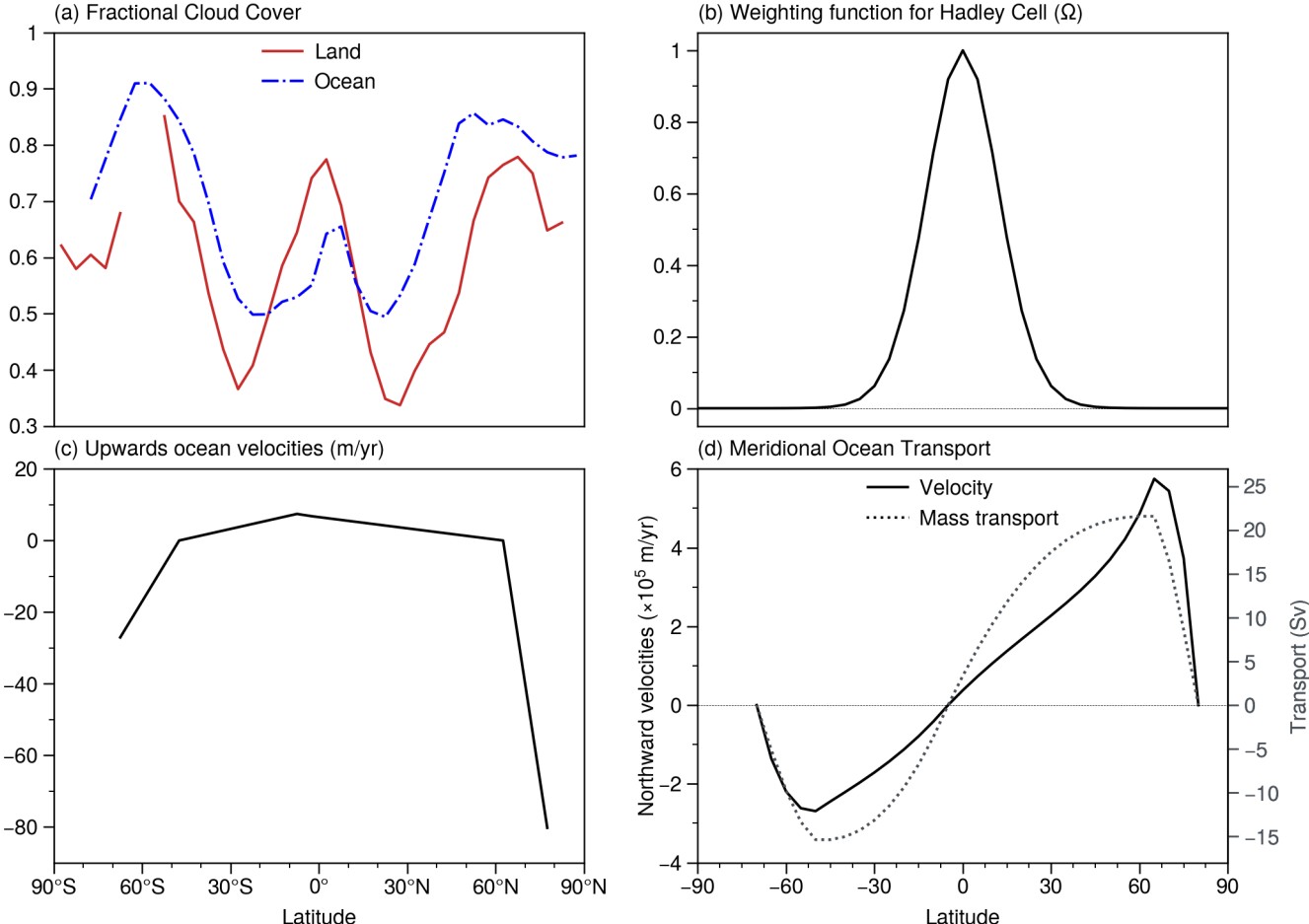

**Figure 1. (a)** Cloud cover fractions over land and ocean taken from pre-industrial simulations of NorESM2 (Seland et al., 2020) for calculations of the shortwave and longwave radiative fluxes. **(b)** The weighting function from Siler et al. (2018) which determines the fraction of atmospheric heat transport carried out by the Hadley cell. **(c)** The prescribed vertical ocean velocities for driving ocean circulation, with upwelling from 50° S to 60° N (with an average upwelling rate of $4\,\mathrm{m\,yr^{-1}}$) and downwelling from 70 to 50° S and from 60 to 80° N. **(d)** The resulting northward horizontal velocities in the uppermost ocean layer (solid black line), together with the mass transport (dotted grey line), in sverdrups ($1 \times 10^6\,\mathrm{m^3\,s^{-1}}$).

where $T_\mathrm{a}'$ is the surface air temperature corrected for the zonal-mean elevation with a global mean lapse rate of $-6.5\,\mathrm{K\,km^{-1}}$. In the current version of the model, this zonal-mean elevation is prescribed (see Sect. 2.4 and Table 2). In the future, we intend to make the zonal-mean elevation dependent on a coupled ice sheet model. The expression for $f_\mathrm{sf}$ is taken from Harvey (1988) as the fractional area of a grid box over which precipitation falls as snow, based on meteorological station data. Therefore, rather than assuming a uniform distribution of snowfall across each grid box, this parameterization allows for only a portion of the land or ocean surface to be snow-covered. As precipitation is assumed to fall uniformly over each grid box, however, this geographic fraction also represents the overall proportion of precipitation that is converted into snow.

### 2.1.4 Atmospheric heat transport

The division of the model into land and ocean raises the question of how to parameterize the "zonal mixing" of air belonging to the same latitudes but overlying each surface type. We adopt an "infinite wind" mixing scenario (Peng et al., 1987; Bintanja, 1997), in which atmospheric temperatures and humidities over land and ocean are both set equal to the zonal mean ($\overline{T_\mathrm{a}}$ and $\overline{Q_\mathrm{a}}$, respectively) at the end of each model time step, according to the land fraction at each grid cell. The assumption of infinite wind mixing between atmosphere over land and ocean is based on previous studies that find that a more realistic representation of zonal mixing produces small differences in model sensitivity (Harvey and Schneider, 1985; Thompson and Schneider, 1979). In other words, one atmospheric layer effectively overlies the entire surface. Following the zonal mixing of temperature and humidity,

meridional atmospheric heat transport is parameterized as a diffusive process along horizontal gradients in moist static energy ($\overline{m}$), expressed as $\overline{m} = c_a \overline{T_a} + L_v \overline{Q_a}$, where $c_a \overline{T_a}$ and $L_v \overline{Q_a}$ represent the dry static and moist components of atmospheric heat content, respectively. Moist static EBMs behave differently to classic (dry) EBMs (Hwang and Frierson, 2010; Feldl and Merlis, 2021) and have shown success at emulating the response of more comprehensive GCMs to climate forcings (Hwang and Frierson, 2010; Roe et al., 2015). Consequently, the total northward transport of energy ($F_{\text{total}}$) within the atmospheric layer is described as follows:

$$F_{\text{total}} = 2\pi R_e \cos(\theta) \rho_a H_a D_a \frac{\delta \overline{m}}{R_e \delta \theta}, \tag{12}$$

where $D_a$ is the atmospheric diffusion coefficient. The total atmospheric heat transport is divided into the dry static ($F_T$) and latent component ($F_Q$). $F_T$ contributes directly to the heating of the atmospheric layer, whereas $F_Q$ transports water vapour within the simplified hydrological cycle (Eq. 8), which is translated into heating of the atmospheric column once precipitation occurs. To capture the equatorward transport of latent heat in the tropics, against meridional gradients in $\overline{m}$, the Hadley cell parameterization introduced by Siler et al. (2018) is included to obtain a more realistic representation of the hydrological cycle. In this parameterization, $F_{\text{total}}$ is partitioned into a Hadley cell ($F_{\text{HC}}$) and an eddy component ($F_{\text{eddy}}$) (Siler et al., 2018):

$$\begin{aligned} F_{\text{total}} &= F_{\text{HC}} + F_{\text{eddy}} \\ &= F_{\text{total}} \, \Omega + F_{\text{total}}[1 - \Omega], \end{aligned} \tag{13}$$

where $\Omega$ is a weighting function (shown in Fig. 1b) that ensures the Hadley cell dominates heat transport in the tropics, whereas eddies control the poleward flux of $\overline{m}$ in the mid-latitudes and polar regions. Where $\Omega = 0$, eddies account for all heat transport via the down-gradient diffusion of both dry static ($F_{T\_\text{eddy}}$) and latent ($F_{Q\_\text{eddy}}$) heat:

$$F_{T\_\text{eddy}} = 2\pi R_e \cos(\theta) \rho_a H_a c_a D_a \frac{\delta \overline{T_a}}{R_e \delta \theta} [1 - \Omega], \tag{14}$$

$$F_{Q\_\text{eddy}} = 2\pi R_e \cos(\theta) \rho_a H_a L_v D_a \frac{\delta \overline{Q_a}}{R_e \delta \theta} [1 - \Omega]. \tag{15}$$

In the Hadley cell, the poleward flux of $\overline{m}$ in its upper branch slightly exceeds the equatorward flux in its lower branch (Hartmann, 2015). Noting that meridional gradients in $\overline{m}$ are relatively flat within this upper branch, Siler et al. (2018) approximate this difference in $\overline{m}$ between the upper and lower branch as $g = \lambda \cdot \overline{m}_{\text{eq}} - \overline{m}$, where $\overline{m}_{\text{eq}}$ is the near-surface moist static energy at the Equator and $\lambda$ is the fractional increase in moist static energy in the upper branch of the Hadley cell relative to $\overline{m}_{\text{eq}}$. In Siler et al. (2018) $\lambda$ is set to 1.06, but for this study $\lambda = 1.03$ to improve our simulation of precipitation for the pre-industrial period. Consequently, the net poleward transport of heat within the Hadley cell is

given as $F_{\text{HC}} = g \, \psi$, where $\psi$ is the mass transport within either the upper or the lower branch of the Hadley cell, which can be solved using Eq. (13) as $\psi = (F_{\text{total}} \, \Omega)/g$. Assuming latent heat transport is confined to the lower branch, the dry static ($F_{T\_\text{HC}}$) and latent ($F_{Q\_\text{HC}}$) contributions to heat transport within the Hadley cell are parameterized using (Siler et al., 2018)

$$F_{T\_\text{HC}} = \psi[g + L_v \, Q_a], \tag{16}$$

$$F_{Q\_\text{HC}} = -\psi \, L_v \, Q_a. \tag{17}$$

Finally, the Hadley cell and eddy contributions to the total northward transport of dry static ($F_T$) and latent ($F_Q$) heat are simply given by

$$\begin{aligned} F_T &= F_{T\_\text{HC}} + F_{T\_\text{eddy}} \\ &= \psi[g + L_v \, Q_a] + 2\pi R_e \cos(\theta) \rho_a H_a c_a D_a \frac{\delta \overline{T_a}}{R_e \delta \theta} \\ &\quad \times [1 - \Omega], \end{aligned} \tag{18}$$

$$\begin{aligned} F_Q &= F_{Q\_\text{HC}} + F_{Q\_\text{eddy}} \\ &= -\psi \, L_v \, Q_a + 2\pi R_e \cos(\theta) \rho_a H_a L_v D_a \frac{\delta \overline{Q_a}}{R_e \delta \theta} \\ &\quad \times [1 - \Omega], \end{aligned} \tag{19}$$

where the divergence of $F_T$ and $F_Q$ (converted into a moisture flux by dividing by $L_v$) is shown on the right side of Eqs. (1) and (8), respectively.

## 2.2 Land

The energy balance of the land surface is expressed by

$$C_1 \, \rho_1 \, H_1 \, \frac{\delta T_1}{\delta t} = S_1 + I_1 - K_1 - L_v E_1, \tag{20}$$

where $C_1$, $H_1$ and $\rho_1$ are the specific heat, depth and density of the ground layer, respectively; $T_1$ is the land surface temperature, $S_1$ is the absorbed shortwave radiation; $I_1$ is the absorbed longwave radiation; $K_1$ is the upward sensible heat flux; and $E_1$ is the upward evaporation flux over land.

### 2.2.1 Snow mass budget

The model evaluates both the proportion of the land surface covered by the snow ($f_{\text{sc}}$) and the average thickness of the snowpack ($d_{\text{sc}}$). A fractional area, $f_{\text{sf}}$, of the precipitation accumulates as snow at the surface according to the hydrological cycle (Sect. 2.1.3). Similarly to the EBM from Harvey (1988), when $f_{\text{sf}}$ is less than the existing area of the snowpack (i.e. $f_{\text{sf}} < f_{\text{sc}}$), this snowfall gain is redistributed over the larger snow-covered area. Conversely, if $f_{\text{sf}} > f_{\text{sc}}$, the snow-covered area is expanded to match the area of snowfall ($f_{\text{sf}} = f_{\text{sc}}$) with the average thickness of the snowpack, $d_{\text{sc}}$, adjusted to conserve the total mass of snow in each grid cell.

If land surface temperatures ($T_l$) exceed the melting point of snow ($T_K$), $T_l$ is reset to $T_K$ and the excess energy is used to melt the snowpack. This rate of melting ($\dot{M}$) is described by

$$\dot{M} = \begin{cases} 0, & T_l \leq T_K, \\ (T_l - T_K) \cdot \frac{C_l \rho_l H_l}{L_f \rho_{ice}}, & T_l > T_K, \end{cases} \quad (21)$$

where $\rho_{ice}$ represents the density of ice. Surface melting of the snowpack is evenly distributed between reducing the average thickness, $d_{sc}$, and the fractional area of the snowpack, $f_{sc}$. Should $\dot{M}$ exceed $d_{sc}$, excess melt is reconverted into heating of the land surface.

### 2.2.2 Land surface albedo

The albedo of the land is determined by the fraction of the surface that is covered in snow, together with the albedo of the snow-covered and snow-free surfaces. The fractional snow cover, $f_{sc}$, is assumed to be evenly distributed between two surface types: bare ground and land ice. The snow albedo ($\alpha_s$) is expressed as a linear function of surface temperature following Bintanja (1997). TS3

$$\alpha_s = \begin{cases} \alpha_{cs}, & T_l' \leq 263\,\text{K}, \\ \alpha_{cs} + [\alpha_{ws} - \alpha_{cs}] \frac{(T_l' - 263)}{10}, & 263\,\text{K} \leq T_l' \leq 273\,\text{K}, \\ \alpha_{ws}, & T_l' > 273\,\text{K}, \end{cases} \quad (22)$$

where $\alpha_{cs}$ is the maximum (or "cold") snow albedo, $\alpha_{ws}$ is the minimum (or "warm") snow albedo and $T_l'$ is the land surface temperature corrected for zonal-mean elevation. The average albedo over bare ground ($\alpha_g$) and ice ($\alpha_i$) is then calculated as the weighted average of the snow-covered and snow-free region.

$$\alpha_g = f_{sc}\alpha_s + (1 - f_{sc})\alpha_{bg}, \quad (23)$$
$$\alpha_i = f_{sc}\alpha_s + (1 - f_{sc})\alpha_{bi}, \quad (24)$$

where $\alpha_{bg}$ and $\alpha_{bi}$ are the albedo of bare ground and the albedo of ice without snow cover, respectively. The assumption of a uniform $\alpha_{bg}$ albedo overlooks the important influence that different vegetation types have on land albedo. In contrast, Bintanja (1997) divides "ice-free" land into present-day distributions of grass and forest cover, though these proportions are held constant over time. Thus, in both approaches, these potentially significant vegetation feedbacks are excluded from Quaternary climate simulations. While including present-day vegetation distribution could improve pre-industrial simulations of ZEMBA, we see limited added value in doing so for studies of orbitally driven climate change. Nonetheless, we recognize that these simplifications in land albedo may affect the strength of albedo feedbacks over land, which could be explored in future applications of the model. The average albedo over land ($\alpha_l$) is the weighted average of $\alpha_g$ and $\alpha_i$, depending on the fractional area over which land is covered by ice ($f_i$):

$$\alpha_l = f_i \alpha_i + (1 - f_i)\alpha_g. \quad (25)$$

Additionally, the impact of the solar zenith angle, $\theta_z$, on albedo is accounted for by increasing the land albedo for zenith angles greater than $60°$ using the following parameterization (Lefebre et al., 2003):

$$\alpha_l = \begin{cases} \alpha_l + \max\left\{0; 0.32 \cdot \frac{1}{2}\left[\frac{3}{1+4\cos\theta_z} - 1\right]\right\}, & \theta_z \geq 60°, \\ \alpha_l, & \theta_z < 60°, \end{cases} \quad (26)$$

with the constraint that $\theta_z > 80° = 80°$. This adjustment increases land albedo for very high zenith angles in the polar regions.

### 2.3 Ocean

Meridional heat transport by the oceans is represented by the zonally averaged ocean circulation model from Bintanja (1997). Extending from $70°\,\text{S}$ to $80°\,\text{N}$, the ocean model comprises six layers of increasing thickness with depth (with a total depth of 4000 m). To induce thermohaline circulation, prescribed vertical velocities (shown in Fig. 1c) are used to produce regions of upwelling from $50°\,\text{S}$ to $60°\,\text{N}$ and downwelling in the polar regions. These vertical velocities remain constant with depth and drive the poleward flow of water in the uppermost model layer (Fig. 1c) and equatorward flow in the bottom layer, thereby generating a conveyor-belt system of ocean heat transport in each hemisphere. The central point of ocean circulation is placed slightly south of the Equator at $5°\,\text{S}$ to improve temperature estimates in the north polar regions. The model effectively consists of two overturning cells, with an average upwelling rate of $4\,\text{m yr}^{-1}$ in the upwelling regions (from 50 to $5°\,\text{S}$ in the southern cell and from $5°\,\text{S}$ to $60°\,\text{N}$ in the northern cell). The transport of ocean heat via eddies and gyres is represented as a diffusive process, along horizontal gradients in surface ocean temperature. Outside of the ocean circulation basin, where ocean fractions are greater than zero (e.g. from $80°\,\text{S}$ to $90°\,\text{N}$), a passive mixed layer of 100 m depth exchanges radiative and turbulent heat with the atmosphere. The temporal evolution of ocean temperature, $T_o$, is described by (Bintanja, 1997)

$$C_o\,\rho_o\,H_o\frac{\delta T_o}{\delta t} + \frac{1}{2\pi R_e^2 f \cos\theta}\frac{\delta F_{ov}}{\delta\theta} + C_o\,\rho_o\,H_o\frac{\delta}{\delta z}(w\,T_o) =$$
$$[S_o + I_o - K_o - L_v E_o] + \frac{1}{2\pi R_e^2 f \cos\theta}\frac{F_{eg}}{\delta\theta} +$$
$$\frac{1}{2\pi R_e^2 f \cos\theta}\frac{\delta F_i}{\delta\theta} + C_o\,\rho_o\,H_o\frac{\delta}{\delta z}\left(D_z\frac{\delta T_o}{\delta z}\right), \quad (27)$$

where $C_o$, $\rho_o$ and $H_o$ are the density, specific heat and depth of each ocean layer; $z$ is the vertical coordinate; $f$ is the fractional width of the ocean basin; $F_{ov}$ is the northward advective heat flux in the top and bottom ocean layer due to overturning; $w$ is the prescribed vertical velocity; $S_o$ is absorbed shortwave radiation; $I_o$ is absorbed longwave radiative flux; $K_o$ is the exchange of sensible heat with the atmosphere; $E_o$ is evaporation at the ocean surface; $F_{eg}$ is the northward diffusive heat flux at the ocean surface due to eddies and gyres;

$F_i$ is the northward diffusive heat flux in the ocean interior; and $D_z$ is the coefficient for vertical heat diffusion. The second term on the left side of the equation represents the divergence of horizontal heat advection which pertains to the top and bottom layer of the ocean model, and the third term represents the divergence of vertical heat advection. On the right side, the first to fourth terms represent the radiative and turbulent heat fluxes and the fifth term is the divergence of horizontal heat diffusion (via eddies and gyres), all of which apply to the uppermost ocean layer. The sixth term on the right side – the divergence of horizontal heat diffusion in the ocean interior – applies to every layer except the surface. Finally, the last term on the right side of the equation represents the divergence of vertical heat diffusion. The northward heat flux associated with $F_{ov}$, $F_{eg}$ and $F_i$ is described by

$$F_{ov} = 2\pi R_e f \cos(\theta) \rho_o C_o H_o u T_o, \tag{28}$$

$$F_{eg} = -2\pi R_e f \cos(\theta) \rho_o C_o H_o D_o \frac{\delta T_o}{R_e \delta \theta}, \tag{29}$$

$$F_i = -2\pi R_e f \cos(\theta) \rho_o C_o H_o D_i \frac{\delta T_o}{R_e \delta \theta}, \tag{30}$$

where $u$ is the horizontal ocean velocity, $D_o$ is the diffusion coefficient related to eddy and gyre transport at the surface and $D_i$ is the coefficient related to horizontal diffusion in the ocean interior. The distribution of $u$ is calculated using the continuity equation from Bintanja (1997, p. 9) and is shown in Fig. 1d.

### 2.3.1   Sea ice

The inclusion of sea ice is important for capturing the seasonal range of surface temperatures at the higher latitudes. By reflecting the majority of incoming solar radiation and reducing heat exchange between the atmosphere and ocean surface, the presence of sea ice reduces the effective thermal inertia of the ocean–atmosphere system. Therefore, ZEMBA includes a simple sea ice model to simulate the latitudinal distribution of sea ice and its modification of ocean albedo. The sea ice model does not account for variations in sea ice thickness or sea ice drifting. Sea ice of a prescribed thickness forms or melts when surface ocean temperatures drop below or exceed a critical threshold ($T_{fo}$) – the freezing temperature of seawater. The heat flux available for the formation or melting of sea ice, $Q_{si}$, is governed by (Gildor and Tziperman, 2001)

$$Q_{si} = \frac{C_o \, \rho_o \, H_o \, A_o}{\Delta t} \, (T_{fo} - T_o), \tag{31}$$

where $A_o$ is the surface area of the ocean. The heat flux is converted into changes in sea ice volume, $V_{si}$, as follows:

$$\frac{\delta V_{si}}{\delta t} = \frac{Q_{si}}{\rho_{ice} L_f} + \frac{P_s \cdot A_{si}}{\rho_{ice}}, \tag{32}$$

where the last term on the right side of the equation represents the snowfall contribution to sea ice volume, with $A_{si}$

being the surface area of the sea ice cover. Sea ice volume is then converted into sea ice areal extent by assuming a constant sea ice thickness ($d_{si}$), which is set to 2 m. When sea ice forms, the temperature of the underlying ocean is reset to $T_{fo}$, and when sea ice covers the entire ocean surface (i.e. $A_{si} = A_o$), the temperature of the sea ice can drop below the freezing point, while the upper ocean layer remains at $T_{fo}$. Variations in the temperature of the surface ocean layer (and thereby sea ice) are determined by the surface energy balance, in addition to advective and diffusive ocean heat fluxes within the ocean circulation model.

### 2.3.2   Ocean albedo

The albedo of the open ocean, $\alpha_{op}$, is parameterized as a function of the solar zenith angle, as derived from aircraft observations (Taylor et al., 1996):

$$\alpha_{op} = \frac{0.037}{1.1\theta_z^{1.4} + 0.15}, \tag{33}$$

with the average albedo for the ocean surface, $\alpha_o$, described by

$$\alpha_o = \alpha_{si} f_{si} + \alpha_{op}(1 - f_{si}), \tag{34}$$

where $\alpha_{si}$ is the albedo of the sea ice and $f_{si}$ is the fraction of the ocean surface covered by sea ice.

### 2.4   Numerical details

The model has the option to be simulated at a 1, 2.5 and 5° resolution. All equations are solved using the forward Euler method, with the majority of calculations performed using a 1 d time step. The exception is the atmospheric and ocean heat transport processes, which may be solved using a shorter time step for numerical stability reasons depending on the choice of model resolution. For all experiments described in the subsequent sections, a model resolution of 5° is chosen. Cloud emissivity is set to 1. The percentage of land and ocean cover for each zonal band is taken from the ICE-6G_C dataset (Argus et al., 2014; Peltier et al., 2014), although a mixed ocean layer is assumed to cover the entire surface from 80–90°) and land occupies the whole area poleward of 75°. Values for key model parameters are summarized in Table 1. These are based on values used in previous studies using an EBM which formed the basis of ZEMBA (Bintanja, 1997; Stap et al., 2014) but with small adjustments to improve the simulated pre-industrial zonal-mean temperature. The coefficient for atmospheric heat transport ($D_a$) has been modified in both hemispheres to improve the simulated polar temperature. Additional sensitivity experiments are presented in Appendix A. The choice of pre-industrial cloud cover can have a significant impact on the simulated climate (Appendix A1). Furthermore, sensitivity experiments conducted for all key model parameters reveal that ZEMBA is particularly sensitive to cloud cover parameters (Appendix A2). Notably,

ZEMBA shows a strong sensitivity to the globally averaged cloud optical depth ($\tau$), which has been used as a tuning parameter to adjust the radiation budget to match that of the present day (Bintanja, 1997; Stap et al., 2014). A comprehensive list of all model variables, parameters and constants is provided in Appendix B.

## 3 Results

We present equilibrium simulations of the model for the pre-industrial period (ZEMBA-PI), a doubling of atmospheric $CO_2$ conditions relative to the pre-industrial period (ZEMBA-$2 \times CO_2$), a 2 % increase in solar insolation (ZEMBA-$S_o + 2\%$) and the Last Glacial Maximum (ZEMBA-LGM). The model takes approximately 3000 model years to reach an equilibrium due to the inclusion of the ocean model with a large heat capacity. Key model parameters remain the same between experiments (Table 1), with the exception of the insolation forcing, atmospheric $CO_2$ concentration and land ice extent (Table 2). The following sections describe output from ZEMBA prior to coupling with an interactive ice sheet model, with fixed land ice fractions and zonal-mean elevations over land taken from ICE-6G_C (Argus et al., 2014; Peltier et al., 2014).

### 3.1 Pre-industrial simulation

For the ZEMBA-PI experiment, the model is forced with insolation using the present-day orbital parameters (Laskar et al., 2004) and an atmospheric $CO_2$ concentration of 284 ppm. To evaluate the accuracy of the ZEMBA-PI model output, we compare it against a selection of zonally averaged climate variables from a pre-industrial simulation of the Norwegian Earth System Model version 2 (NorESM2) (Seland et al., 2020). In addition, we compare ZEMBA-PI against the ERA5 atmospheric reanalysis product averaged between 1940 and 1970 (Hersbach et al., 2023), which belongs to a period when $CO_2$ concentrations were $\sim 26$–$45$ ppm higher than the pre-industrial levels but provides one of the earliest observational constraints on the full-field climate. As can be seen in the subsequent figures (Figs. 2–7), zonally averaged climate variables are very similar between NorESM2 and ERA5 1940–1970.

The zonal-mean surface air temperatures are shown in Fig. 2. For annual-mean temperatures (Fig. 2a), the latitudinal structure corresponds nicely between ZEMBA-PI, NorESM2 and ERA 1940–1970. Moreover, the DJF mean (Fig. 2b) and JJA mean (Fig. 2c) demonstrate that the model successfully captures the seasonal amplitude of temperatures at the higher latitudes in accordance with both NorESM2 and ERA5 1940–1970. The globally averaged surface air temperature from ZEMBA-PI is 13.87 °C, compared with 13.78 °C from NorESM2 and 13.72 °C from ERA5 1940–1970 (Table 3). When the ZEMBA-PI experiment is compared only to

NorESM2, the difference in annual-mean temperatures never exceeds $\sim 4$ °C (Fig. 2d). The same is true for ERA5 1940–1970, with the exception of 70–90° N, as ERA5 produces higher temperatures in this region due to much higher winter temperatures (Fig. 2b), which is presumably in response to higher $CO_2$ levels. Between 60 and 30° S, ZEMBA-PI appears to consistently overestimate temperatures, both annually (Fig. 2d) and seasonally (Fig. 2e–f), by as much as $\sim 5$ °C when compared to NorESM2 and ERA5 1940–1970. Between 30 and 60° N, on the other hand, the annual-mean temperatures agree well amongst the models and reanalysis, but ZEMBA-PI seems to underestimate the seasonal range in temperatures at these latitudes. Overall, despite some biases, ZEMBA captures both the annual mean and the seasonal range of surface air temperatures with good accuracy for the pre-industrial period.

Figure 3 shows climate variables related to the hydrological cycle including precipitation, snowfall and evaporation. Given the simplicity of ZEMBA, precipitation rates from the ZEMBA-PI experiment are well captured when compared to NorESM2 and ERA5 1940–1970 (Fig. 3a). The inclusion of the Hadley cell parameterization from Siler et al. (2018) generates a convergence of moisture and strong precipitation rates near the Equator. The largest difference between ZEMBA-PI and the other models and reanalysis is located around the Equator (Fig. 3c), as the precipitation maximum is located at the Equator for ZEMBA-PI, whereas the maxima for NorESM2 and ERA5 1940–1970 are located north of the Equator in accordance with the mean position of the Intertropical Convergence Zone (ITCZ). In the mid-latitudes and polar regions, precipitation rates from ZEMBA-PI are in close agreement (within $1 \, \mathrm{mm \, d^{-1}}$) with those simulated by the complex models. The snowfall rates (Fig. 3) from ZEMBA-PI over the NH correspond nicely to NorESM2 and ERA5 1940–1970. For the Southern Hemisphere (SH), on the other hand, while ZEMBA-PI captures the location of maximum snowfall at 60° S, it appears to underestimate the snowfall rate by half (Fig. 3c). As for evaporation, the zonally and annually averaged fluxes from ZEMBA-PI are very similar to NorESM2 and ERA5 (Fig. 3b), with the differences between them never exceeding $1 \, \mathrm{mm \, d^{-1}}$ (Fig. 3d).

The surface and planetary albedo compares favourably between ZEMBA-PI, NorESM2 and ERA5 1940–1970 (Fig. 4). Most importantly, ZEMBA-PI reproduces the poleward enhancement of albedo at the higher latitudes due to the presence of snow and sea ice cover. The difference in surface and planetary albedo as simulated by ZEMBA-PI in comparison to the other models and reanalysis never exceeds 0.2 (Fig. 4c–d). One limitation of the shortwave parameterization used in ZEMBA is the overestimation of the planetary albedo by $\sim 0.1$ at the polar latitudes (Fig. 4c) as noted by Bintanja (1997). However, as the planetary albedo is slightly underestimated at the mid-latitudes, the globally averaged planetary albedo is very similar between ZEMBA (0.31), NorESM2 (0.32) and ERA5 1940–1970 (0.31) (Table 3). At

**Table 1.** Selection of important parameters used in the atmospheric, land and ocean components of the model.

| Parameter | Units | Value | Description |
|---|---|---|---|
| **Atmosphere** | | | |
| $\tau$ | – | 3.0 | Cloud optical depth |
| $\kappa_l$ | $m\,s^{-1}$ | 0.01 | Turbulent heat flux coefficient over land |
| $\kappa_o$ | $m\,s^{-1}$ | 0.006 | Turbulent heat flux coefficient over ocean |
| $r_{max}$ | – | 80 | Maximum relative humidity |
| $D_a$ | $m\,s^{-1}$ | $0.7 \times 10^6$ (SH); $0.84 \times 10^6$ (NH) | Diffusion coefficient for atmospheric heat transport |
| **Land** | | | |
| $\alpha_g$ | – | 0.15 | Albedo of bare ground |
| $\alpha_{cs}$ | – | 0.8 | Albedo of cold ("dry") snow |
| $\alpha_{ws}$ | – | 0.4 | Albedo of warm ("wet") snow |
| $\alpha_i$ | – | 0.8 | Albedo of land ice |
| **Ocean** | | | |
| $\alpha_{si}$ | – | 0.7 | Albedo of sea ice |
| $D_o$ | $m\,yr^{-1}$ | $5 \times 10^{10}$ | Diffusion coefficient for horizontal heat transport at surface |
| $D_i$ | $m\,yr^{-1}$ | $1.5 \times 10^{10}$ | Diffusion coefficient for horizontal heat transport in ocean interior |
| $D_z$ | $m\,yr^{-1}$ | $5 \times 10^3$ | Diffusion coefficient for vertical heat transport |
| $d_{si}$ | m | 2 | Sea ice thickness |

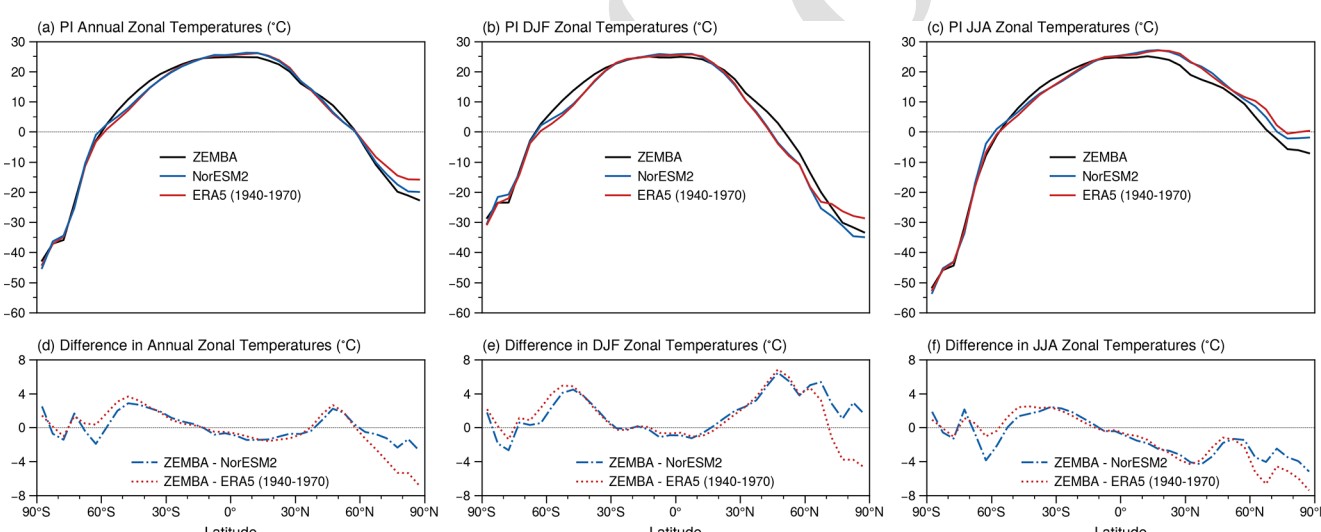

**Figure 2. (a–c)** The annual **(a)**, December–January–February **(b)** and June–July–August **(c)** average of zonal-mean surface air temperature for the pre-industrial period (PI), as simulated by ZEMBA (black lines) in comparison to NorESM2 (blue lines) and the ERA5 climatology from 1940 to 1970 (red lines). **(d–f)** The difference between ZEMBA and the other models and observations.

the surface, the onset of higher surface albedo agrees well with NorESM2 and ERA5 1940–1970 in both hemispheres. Overall, the global mean surface albedo for ZEMBA-PI is 0.15, which is in agreement with both NorESM2 and ERA5 1940–1970 (Table 3).

Figure 5 shows the seasonal cycle in the areal extent of snow cover over land and sea ice coverage. Over land in the NH (Fig. 5a), the ZEMBA-PI experiment underestimates

the winter maximum in snow coverage by $> 1 \times 10^{13}\,m^2$ and thereby simulates a smaller seasonal amplitude in snow cover when compared to NorESM2 and ERA5 1940–1970 reanalysis. As for the SH (Fig. 5a), all models and reanalysis have similar areal extents in snow cover and produce negligible seasonal variations. The ZEMBA-PI experiment shows most success in simulating sea ice cover in the NH (Fig. 5b), capturing both the amplitude and the phase of sea

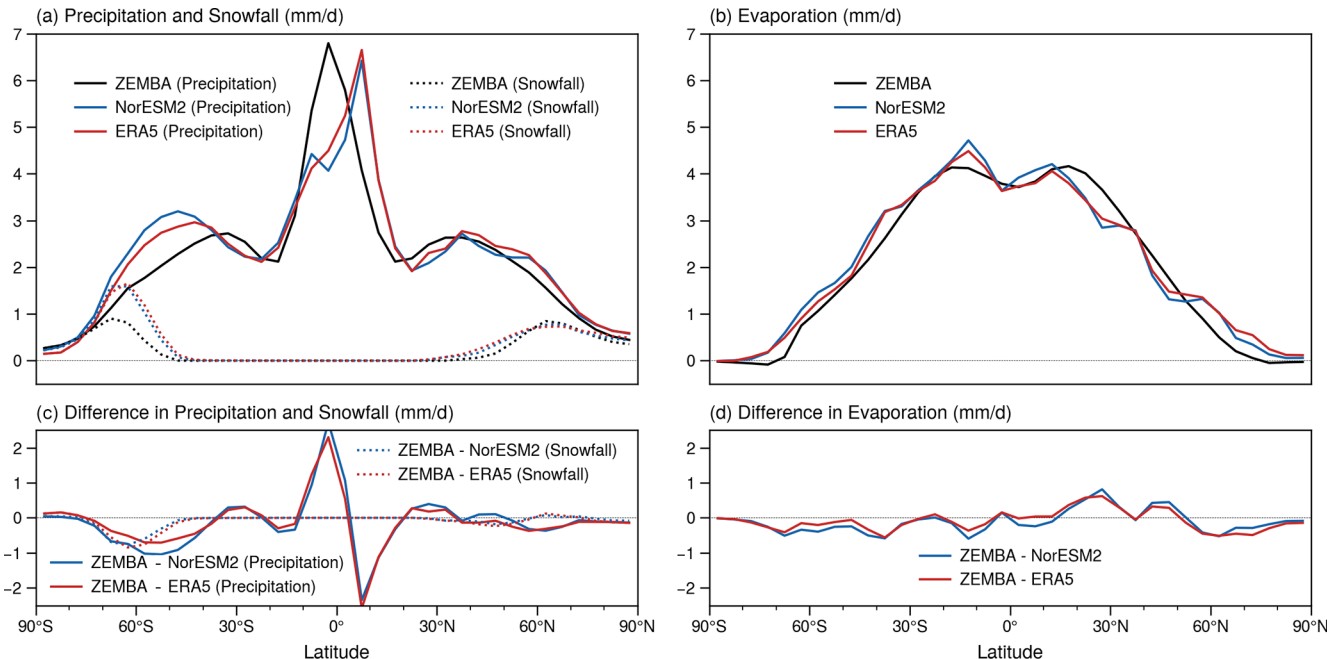

**Figure 3. (a–b)** The annual- and zonal-mean rates of precipitation and snowfall **(a)** and evaporation **(b)**, as simulated by ZEMBA (black lines) in comparison to NorESM2 (blue lines) and the ERA5 1940–1970 climatology (red lines). Precipitation is shown in solid lines, and snowfall is shown in dotted lines **(a, c)**. **(c–d)** The difference between ZEMBA and the other models and reanalysis.

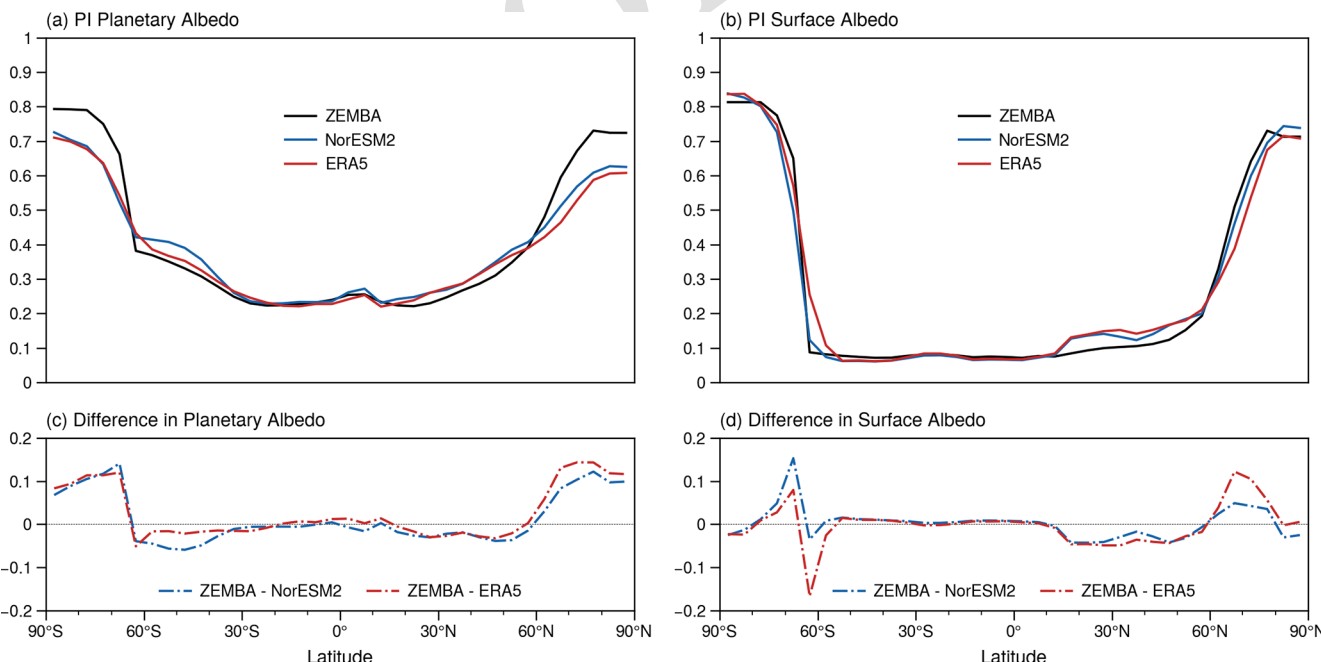

**Figure 4. (a–b)** The annual- and zonal-mean planetary **(a)** and surface **(b)** albedo for the pre-industrial period (PI), as simulated by ZEMBA (black lines) in comparison to NorESM2 (blue lines) and the ERA5 climatology from 1940 to 1970 (red lines). **(c–d)** The difference between ZEMBA and the other models and observations.

**Table 2.** Boundary conditions for ZEMBA experiments. PD denotes present day.

| Experiment | Insolation | Elevation/ice | $CO_2$ |
|---|---|---|---|
| ZEMBA-PI | PD | PD | 284 ppm |
| ZEMBA-LGM | 21 ka | 21 ka | 184 ppm |
| ZEMBA-$2 \times CO_2$ | PD | PD | 568 ppm |
| ZEMBA-$S_o + 2\%$ | PD + 2 % | PD | 284 ppm |

ice cover changes in reference to NorESM2 and ERA5 1940–1970 (Fig. 5b). For sea ice cover in the SH (Fig. 5b), however, the ZEMBA-PI experiment underestimates the seasonal amplitude.

The annual- and zonal-mean radiative fluxes exchanged at the TOA and the surface are shown in Fig. 6. For the TOA, the amount of absorbed shortwave radiation (ASR) estimated using the shortwave parameterization from ZEMBA (see Sect. 2.1.1) compares favourably with NorESM2 and ERA5 1940–1970, albeit with a slight underestimation of ASR in the polar latitudes by $\sim 10$–$20\,\mathrm{W\,m^{-2}}$ (Fig. 6a). The largest differences between ZEMBA-PI and the other models and reanalysis reside in the outgoing longwave flux (OLR) at the TOA (Fig. 6a). In particular, there is a pronounced overestimation of OLR in the tropics (of up to $\sim 30\,\mathrm{W\,m^{-2}}$) and a pronounced underestimation of OLR in the polar latitudes, especially in the Northern Hemisphere (of up to $50\,\mathrm{W\,m^{-2}}$). Consequently, these differences in OLR are reflected in the net radiative flux (NET) received at the TOA, with ZEMBA-PI receiving $\sim 25\,\mathrm{W\,m^{-2}}$ less NET energy at the Equator and up to $\sim 35\,\mathrm{W\,m^{-2}}$ more NET energy in the high northern latitudes (Fig. 6c). As for the surface radiative fluxes (Fig. 6b), the ASR, OLR and NET radiative fluxes are quite similar but tend to be slightly underestimated in ZEMBA-PI when compared to the other datasets, with the difference in the NET fluxes never exceeding $\sim 25\,\mathrm{W\,m^{-2}}$ (Fig. 6d). Overall, despite some clear discrepancies in the ASR and OLR, the ZEMBA-PI experiment produces net radiative fluxes at the TOA and the surface which compare favourably with more complex models, generally falling within $\sim 25\,\mathrm{W\,m^{-2}}$ of those simulated by NorESM2 and ERA5 (Fig. 6c–d).

The simulated northward heat transport via the atmosphere and ocean is depicted in Fig. 7 in reference to NorESM2. We note that NorESM2 heat transport values replicate those estimated from 2000 to 2014 using ERA-Interim reanalysis (Trenberth and Fasullo, 2017), including total heat transport exceeding 5.5 PW in each hemisphere and ocean heat transport peaking at around 2 PW at $15°$ N. The total heat transport in ZEMBA-PI is lower than in NorESM2 in each hemisphere (Fig. 7a) because of reduced heat transport in the atmospheric layer (Fig. 7b). Ocean heat transport corresponds nicely to that inferred from NorESM2 (Fig. 7b), with a maximum value of $\sim 1.5$ and $\sim 1$ PW in the NH and SH, respectively. However, the location of maximum ocean heat trans-

port is located at $\sim 30°$ N/S in the ZEMBA-PI experiment compared to $\sim 15°$ N/S for NorESM2. While the maximum atmospheric heat transport is underestimated in ZEMBA-PI, the location of maximum heat transport corresponds nicely to NorESM2 at $\sim 45°$ N/S. For the ZEMBA-PI experiment, latent heat transport contributes significantly to the poleward flux of atmospheric heat at the mid-latitudes, whereas dry static heat transport dominates the polar regions, in keeping with NorESM2 (Fig. 7c). Moreover, dry static transport peaks at $\sim 15°$ N/S and latent heat transport peaks at $\sim 40°$ N/S for both ZEMBA-PI and NorESM2. The inclusion of the Hadley cell parameterization (Siler et al., 2018) successfully produces the equatorward flux of latent heat seen in the tropics. Overall, while dry static and latent heat transport in the atmosphere is underestimated in the ZEMBA-PI experiment compared to NorESM2, the overall configuration of atmospheric heat transport is very similar between the models.

## 3.2  2xCO$_2$ and +2 % insolation

A common method to test the sensitivity of climate models is to impose changes in atmospheric $CO_2$ concentrations or solar insolation ($S_0$). Therefore, the temperature response to a doubling of atmospheric $CO_2$ concentrations (ZEMBA-$2 \times CO_2$) and a 2 % increase in the solar constant (ZEMBA-$S_o + 2\%$) is shown in Fig. 8, keeping all other boundary conditions the same as in the ZEMBA-PI experiment. It should be noted that land ice fractions and elevations are kept fixed for these experiments. For a doubling of atmospheric $CO_2$ concentrations, the global mean temperature is 3.6 °C higher. The most notable feature is that the NH is significantly more sensitive than the SH, reaching an annual-mean rise of $> 15$ °C in the high northern latitudes, compared to $< 8$ °C over Antarctica (Fig. 8a). In addition, temperature changes are strongest during the winter months of both hemispheres (Fig. 8b). The response to a 2 % increase in the solar constant is very similar to a doubling of atmospheric $CO_2$ concentrations, with an equivalent rise in global mean surface air temperature of 3.3 °C but with more muted warming over Antarctica. In reference to other works, comparisons are made complicated by the fact that GCM simulations involving a doubling or quadrupling of atmospheric $CO_2$ are often not run to their equilibrium climate state due to considerable computing times. However, the global mean warming from the ZEMBA-$2 \times CO_2$ experiment (3.6 °C), known as the equilibrium climate sensitivity (ECS), fits comfortably within the "likely" range of 1.5–4.5 °C estimated in the Fifth Assessment Report of the Intergovernmental Panel on Climate Change (Collins et al., 2013). In addition, polar amplification in surface warming that is strongest in the NH and during the winter months is consistent with both historical observations (1979–2014) and GCMs responding to an abrupt quadrupling of the atmospheric $CO_2$ concentration (Hahn et al., 2021).

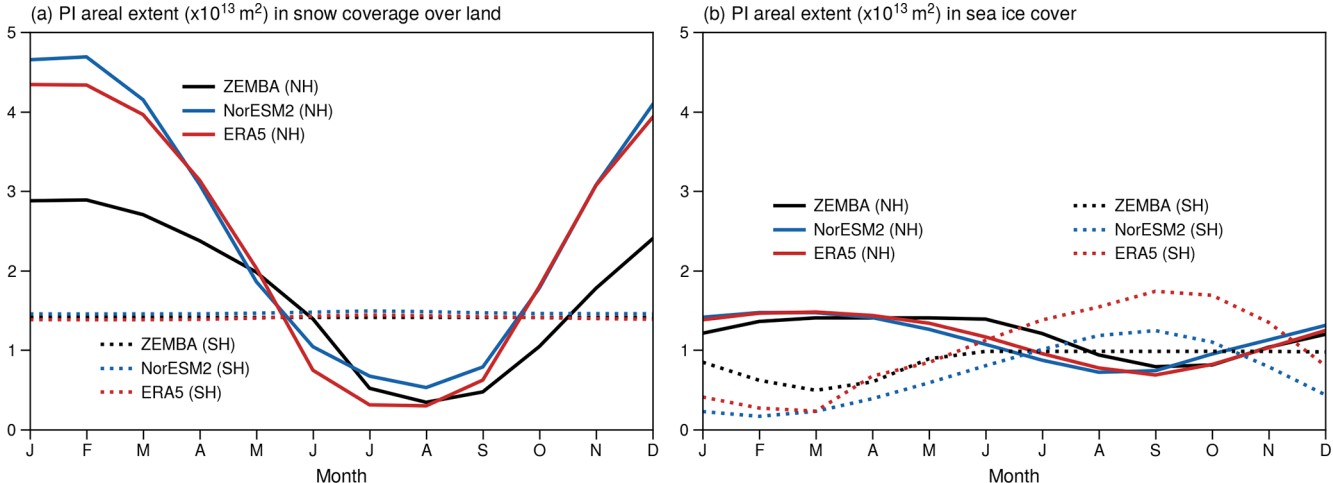

**Figure 5.** Monthly variations in the areal extent of snow coverage over land **(a)** and sea ice coverage over the ocean **(b)** for the pre-industrial period, as simulated by ZEMBA (black lines) in comparison to NorESM2 (blue lines) and the ERA5 1940–1970 climatology (red lines) for the Northern Hemisphere (solid lines) and the Southern Hemisphere (dotted lines). For snow cover over land, monthly averaged ERA5 reanalysis is taken over a shorter period from 1950 to 1970 (Muñoz Sabater, 2019) due to data availability.

**Table 3.** Selection of annual-mean variables from a pre-industrial simulation of ZEMBA in comparison to NorESM2 and ERA5 1940–1970 climatology.

| Variable | ZEMBA | NorESM2 | ERA5 (1940–1970) |
|---|---|---|---|
| Global mean surface air temperature (°C) | 13.87 | 13.78 | 13.72 |
| Global mean planetary albedo | 0.31 | 0.32 | 0.31 |
| Global mean surface albedo | 0.15 | 0.15 | 0.16 |
| Snow cover ($\times 10^{13}$ m$^2$) | NH: 1.72; SH: 1.41 | NH: 2.52; SH: 1.46 | NH: 2.37; SH: 1.40 |
| Sea ice cover ($\times 10^{13}$ m$^2$) | NH: 1.18; SH: 0.86 | NH: 1.15; SH: 0.68 | NH: 1.14; SH: 1.01 |
| Global mean precipitation rate (mm d$^{-1}$) | 2.77 | 2.85 | 2.86 |
| Global mean snowfall rate (mm d$^{-1}$) | 0.14 | 0.21 | 0.22 |
| Peak atmospheric heat transport (PW) | NH: 3.42; SH: 2.84 | NH: 4.66; SH: 4.97 | – |
| Peak ocean heat transport (PW) | NH: 1.62; SH: 0.99 | NH: 1.63; SH: 0.84 | – |

## 3.3 Last Glacial Maximum

To test the ability of ZEMBA to simulate climates other than the pre-industrial period, we perform a simulation of the LGM (hereafter the ZEMBA-LGM experiment). The LGM remains a natural focus for climate models constraining their climate sensitivity because it was the most recent cold extreme when atmospheric $CO_2$ concentrations were $\sim 100$ ppm lower than in the pre-industrial period (Bereiter et al., 2015) and continental ice sheets reached their maximum extent over North America and Fennoscandia. For LGM boundary conditions, the model is forced with insolation using the orbital parameters from 21 ka (Laskar et al., 2004), prescribed changes in land elevation and land ice fractions from ICE-6G_C (Argus et al., 2014; Peltier et al., 2014), and an atmospheric $CO_2$ concentration of 184 ppm (Bereiter et al., 2015) (see Table 2). The differences between the LGM and PI climates, as simulated by ZEMBA, are compared to an ensemble of state-of-the-art climate mod-

els which contributed to PMIP3 and PMIP4 compiled by Kageyama et al. (2021). In addition, we compare our simulation to a recent LGM reconstruction from Annan et al. (2022) which combines an ensemble of climate model simulations with proxy-based estimates of surface temperature using a data assimilation approach. In the ZEMBA-LGM experiment shown in Fig. 9, there are no changes to the strength or the configuration of ocean circulation.

The changes in annual- and zonal-mean surface air temperatures for the ZEMBA-LGM simulation are shown in Fig. 9a. The temperature decrease around the tropics (30° S–30° N) of $-1.95$ °C is comparable to the other reconstructions but falls slightly on the lower end of estimated cooling (Table 4). In the extratropics, however, ZEMBA appears to generate cooling which is slightly too strong in the NH and too mild in the SH when compared to the multi-model averages from PMIP3, PMIP4 and the Annan et al. (2022) reconstruction. The latitudinal distribution of cooling always falls

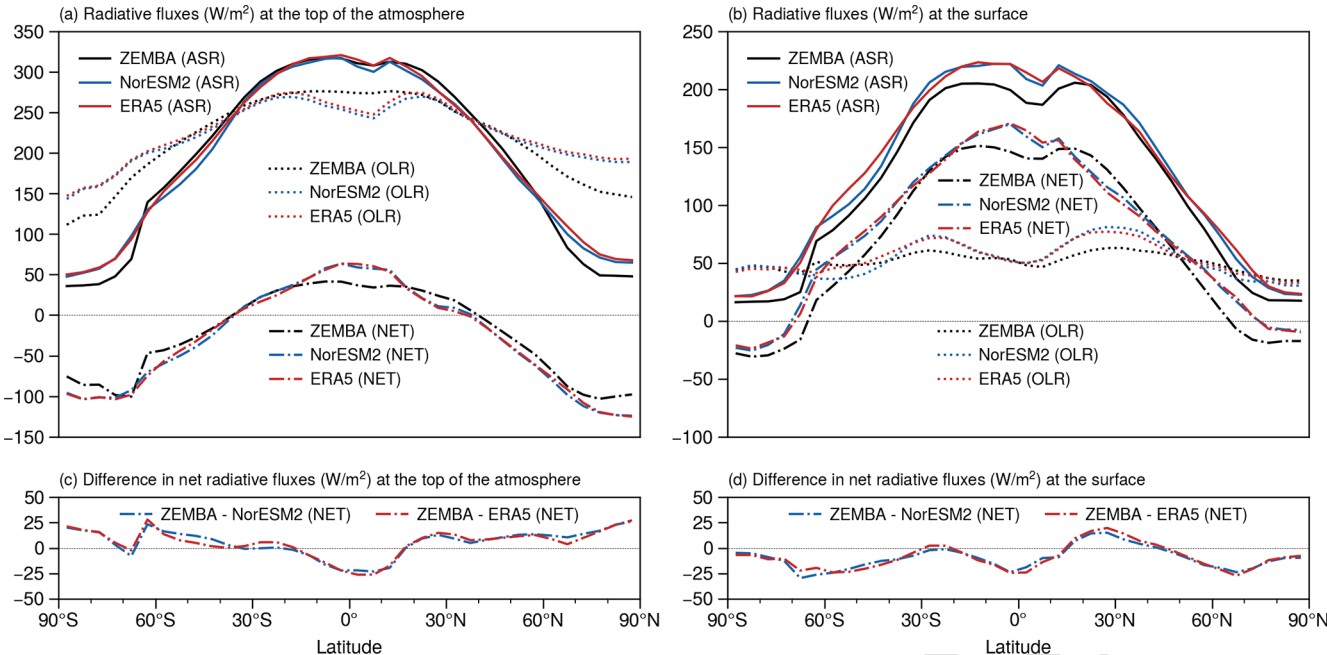

**Figure 6. (a–b)** The annual- and zonal-mean radiative fluxes at the top of the atmosphere **(a)** and the surface **(b)**, as simulated by ZEMBA (black lines) in comparison to NorESM2 (blue lines) and the ERA5 1940–1970 climatology (red lines). Shown are the absorbed shortwave radiation (ASR: solid lines), outgoing longwave radiation (OLR: dotted lines) and net radiation (NET: dash-dotted lines) radiative fluxes. The outgoing longwave radiation at the surface refers to net upward longwave radiation, i.e. the upward longwave radiative flux at the surface minus the downward longwave radiative flux at the bottom of the atmospheric layer. **(c–d)** The difference in the net radiative flux between ZEMBA and the other models and reanalysis.

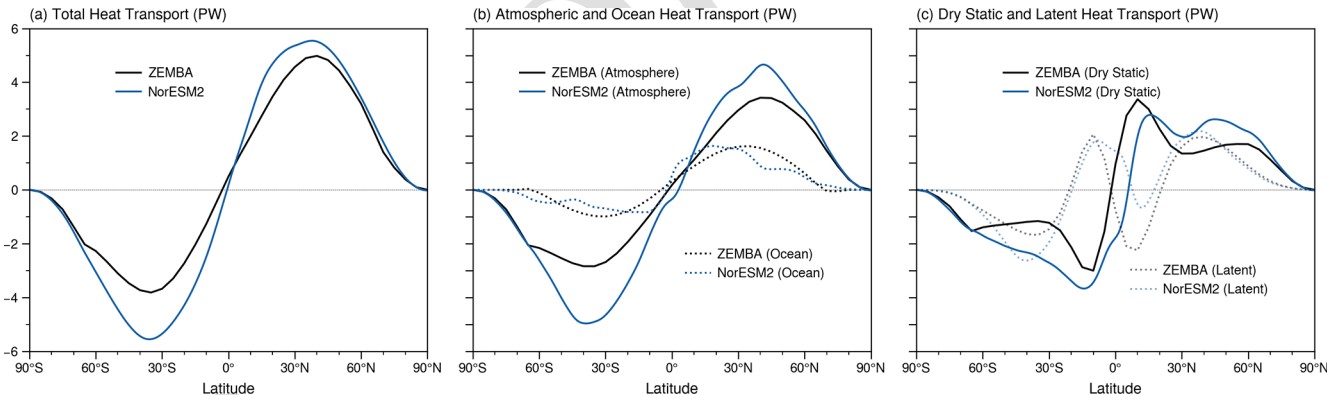

**Figure 7.** The northward transport of atmospheric and ocean heat in the pre-industrial period, as simulated by ZEMBA (black lines) in comparison to NorESM2 (blue lines). Shown is the total heat transport **(a)**, the atmospheric and ocean components **(b)**, and the partition of atmospheric heat transport into the dry static and latent components **(c)**. For NorESM2, annual-mean heat transport is inferred by assuming the system is in equilibrium and the heat transport is equal to energy imbalance at any latitude. More specifically, the energy imbalance at the top of the atmosphere (for total heat transport), in the atmosphere (for atmospheric heat transport) or at the surface (for ocean heat transport) is integrated from the South Pole.

within the range simulated across the PMIP3–PMIP4 model ensemble (yellow-shaded area, Fig. 9), but this is not always the case at the higher latitudes for the Annan et al. (2022) reconstruction (blue-shaded area, Fig. 9), which is constrained by proxies for surface temperatures. Overall, despite some discrepancies in the higher latitudes, the global mean cooling

from the ZEMBA-LGM experiment is −4.11 °C compared to −4.71 °C from PMIP3, −4.77 °C from PMIP4 and −4.46 °C from Annan et al. (2022) (Table 4). For LGM − PI precipitation rates (Fig. 9b), ZEMBA captures the widespread decrease in precipitation, which again falls within the range estimated by the PMIP3–PMIP4 model ensemble. The re-

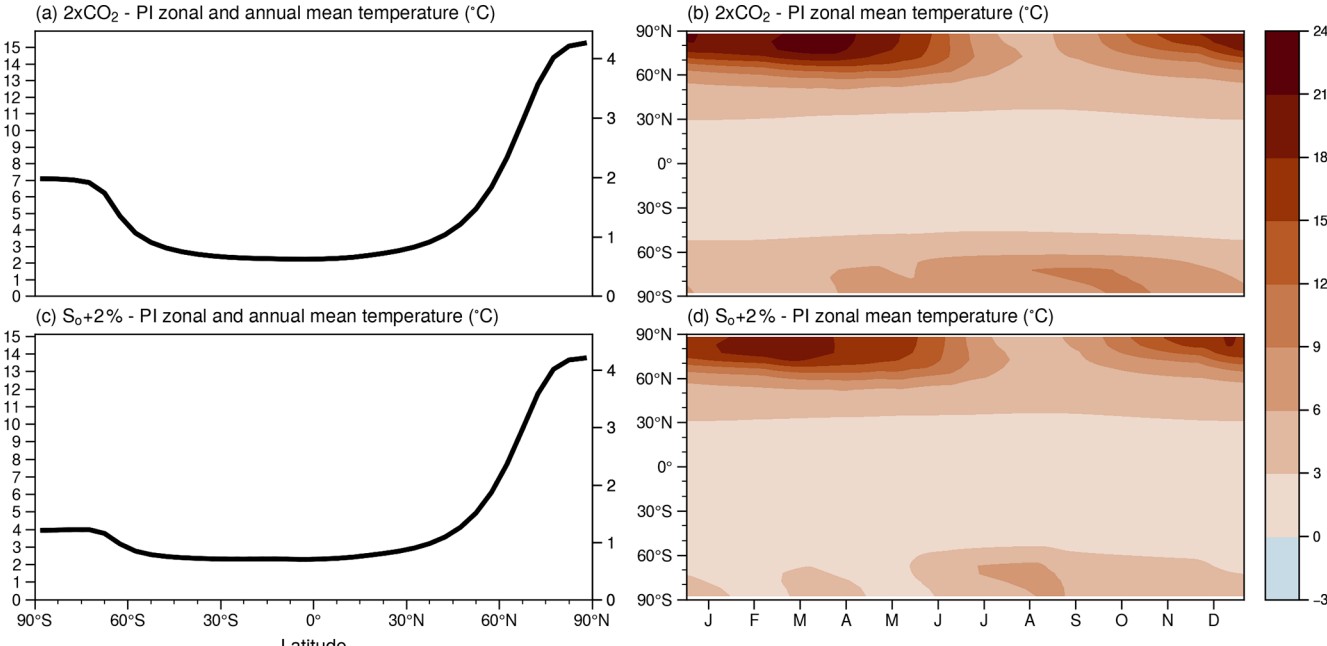

**Figure 8.** Changes in surface air temperature (relative to the PI period) following a doubling of the atmospheric $CO_2$ concentration from 284 to 568 ppm (**a**, **b**; 2x$CO_2$) and a 2 % increase in the solar constant (**c**, **d**; $S_o + 2$ %). (**a, c**) Changes in the annual-mean temperature, which are also shown normalized by the global mean warming (on the right-hand side). (**b, d**) Changes in the seasonal cycle in temperature.

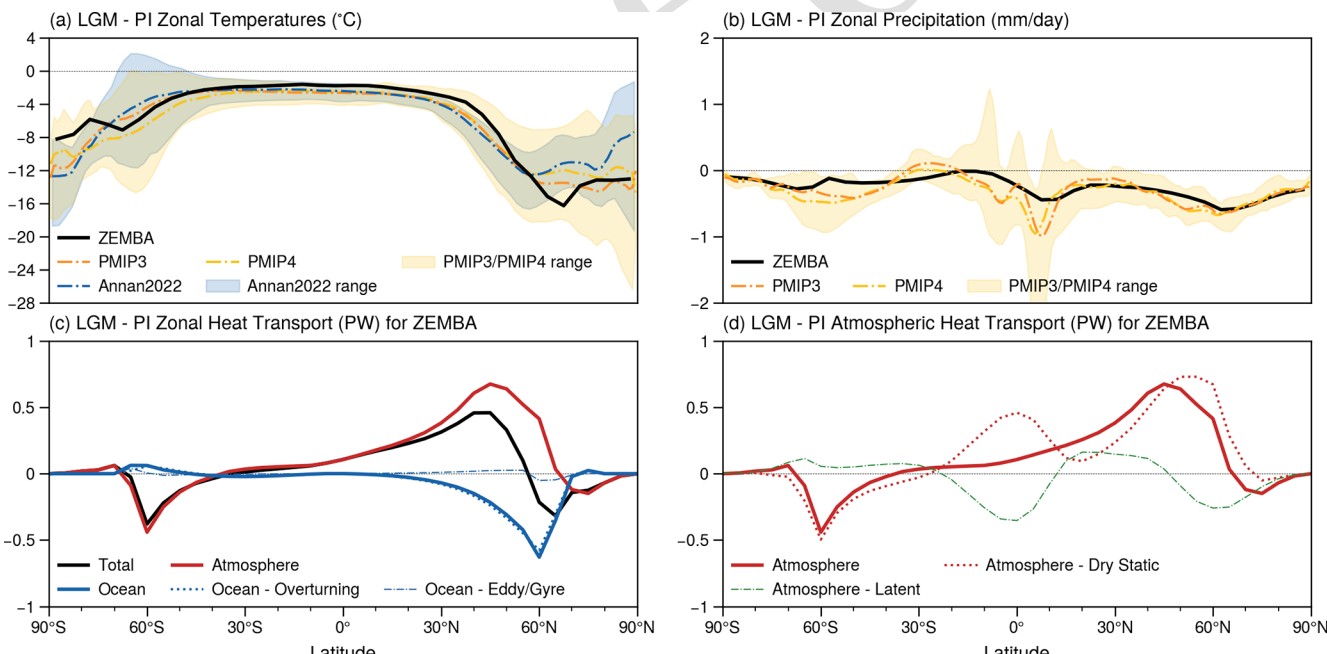

**Figure 9. (a, b)** The LGM minus PI (LGM − PI) surface air temperatures **(a)** and precipitation **(b)** as simulated by ZEMBA (solid black line) in comparison to the multi-model mean from PMIP3 (dash-dotted orange line) and PMIP4 (dash-dotted yellow line) and (in panel **a** only) the data assimilation products from Annan et al. (2022) (dash-dotted blue line). The yellow- and blue-shaded areas represent the range of LGM − PI surface air temperatures and precipitation as reconstructed by PMIP3–PMIP4 and Annan et al. (2022), respectively. **(c, d)** The differences in northward heat transport between the LGM and PI experiments for ZEMBA, including total heat transport (solid black line), atmospheric heat transport (solid red line), ocean heat transport (solid blue line), the decomposition of ocean heat transport into ocean overturning (dotted blue line) and eddy and gyre transport (dashed blue line) **(c)** and the decomposition of atmospheric heat transport (solid red line) into dry static (dotted red line) and latent (dashed green line) heat transport **(d)**.

ductions are largest in the NH extratropics, similarly to surface air temperatures. However, there are a number of discrepancies between ZEMBA LGM − PI precipitation rates and the PMIP3 and PMIP4 model ensembles, such as at 5–10° N, where the PMIP ensemble generates a much stronger reduction in precipitation, and at 30–10° S, where the PMIP ensemble simulates higher zonal-mean precipitation in the LGM due to increased precipitation in the subtropical Pacific Ocean (Kageyama et al., 2021). Overall, the global mean decrease in precipitation is −0.24 mm d$^{-1}$ for ZEMBA compared to −0.29 mm d$^{-1}$ in PMIP3 and −0.33 mm d$^{-1}$ in PMIP4 (Table 5).

The differences in northward heat transport between the ZEMBA-LGM and ZEMBA-PI experiments are shown in Fig. 9c–d. Starting with total heat transport (Fig. 9c), the most notable feature is a significant peak in northward heat transport (of ∼0.5 PW) at 45° N, followed by a significant trough (of ∼0.3 PW) at 60° N. The peak is associated with greater atmospheric heat transport in the LGM experiment (Fig. 9c–d), which is almost entirely because of larger fluxes of dry static energy in the atmospheric layer (Fig. 9d). The reduced northward heat transport at 60° N, on the other hand, is due to a significant 0.5 PW decrease in ocean heat transport (Fig. 9c), associated with sea ice expansion in the LGM experiment. As the surface ocean layer underlying sea ice rests at the freezing point of seawater (see Sect. 2.3.1), there is effectively zero meridional ocean heat fluxes at latitudes covered by sea ice. Consequently, sea ice expansion is accompanied by large drops in ocean heat transport, via both overturning and eddy and gyre transport, at latitudes which were previously free of sea ice in the PI experiment.

In addition to the standard ZEMBA-LGM simulation, we recreate the sensitivity experiment from Bintanja and Oerlemans (1996) by progressively adding each of the LGM boundary conditions (shown in Table 2) followed by changes to the strength of ocean circulation (Fig. 10). Using the EBM from Bintanja (1997), Stap et al. (2014) also found that introducing a mechanism to shift the mid-point of ocean circulation further south during glacial conditions was necessary to produce surface temperatures in closer agreement with observation. Therefore, we additionally investigate the impact of changing the mid-point of ocean circulation from 5 to 15° S. To summarize, we progressively add each of the following:

- *ice* – LGM land ice fractions and zonal-mean land elevations;

- *CO₂* – LGM $CO_2$ concentrations;

- *Inso* – LGM insolation forcing, which is the same as in the standard ZEMBA-LGM experiment;

- *Oc: 15° S* – a shift in the mid-point of ocean circulation from 5 to 15° S;

- *75 % Ov* – a 25 % reduction in the strength of ocean overturning;

- *50 % Ov* – another 25 % (50 % total) reduction in the strength of ocean overturning.

The changes in temperature and precipitation – averaged over the tropics, extratropics and globally – following the addition of each boundary condition are shown in Tables 4 and 5, respectively. Firstly, the addition of LGM land ice fractions and elevation causes a large decrease in temperature (−5.06 °C) and precipitation (−0.22 mm d$^{-1}$) in the NH extratropics, whereas the effects in the tropics and the SH extratropics are minimal. A subsequent reduction in atmospheric $CO_2$ concentrations generates widespread cooling, drying and a further reduction in global mean temperature of −2.08 °C, the latter of which is stronger than the initial cooling caused by the LGM ice sheets (−1.69 °C). The subsequent addition of the LGM insolation forcing causes a further decrease in global mean temperature of 0.34 °C. As in Stap et al. (2014), a shift in the mid-point of ocean circulation to 15° S produces surface temperatures which are in better agreement with LGM reconstructions. In particular, shifting the ocean circulation mid-point 10° further south decreases temperature and precipitation rates in the SH extratropics and increases temperature and precipitation in the NH extratropics, with negligible changes in global mean temperature and precipitation. Finally, reductions in the strength of overturning circulation cause simultaneous warming in the tropics and cooling in the higher latitudes of each hemisphere, which generate global mean cooling and a larger disagreement with the PMIP3–PMIP4 ensemble and LGM reconstruction from Annan et al. (2022).

## 4 Discussion

Our simulation of the pre-industrial period demonstrates that ZEMBA can describe the zonally averaged climate of this period with reasonable accuracy. Surface air temperatures are in strong agreement with NorESM2 and ERA5 1940–1970 both annually and seasonally (Fig. 2a–c), generally falling within 4 °C of those estimated by NorESM2 (Fig. 2d–f). Most notably, given the simplicity of ZEMBA, the model shows success in emulating precipitation and snowfall rates for the pre-industrial climate (Fig. 3a), particularly at the polar latitudes. Furthermore, the model captures the poleward enhancement of the surface and planetary albedo due to the presence of snow and sea ice cover (Fig. 4), with the seasonal range of sea ice cover in the NH corresponding nicely to both NorESM2 and ERA5, varying between ∼0.8 and ∼1.4 × 10$^{13}$ m² (Fig. 5b). The ocean transport model shows success in emulating zonally averaged ocean heat transport, which makes a significant contribution to total heat transport at the lower latitudes, peaking with values of around 1 and 1.5 PW for the SH and NH, respectively (Fig. 7b). Similarly, the model predicts that atmospheric heat transport dominates in the middle to high latitudes and reaches its maximum levels at around ∼45° in each hemisphere. Finally, looking specifically at the

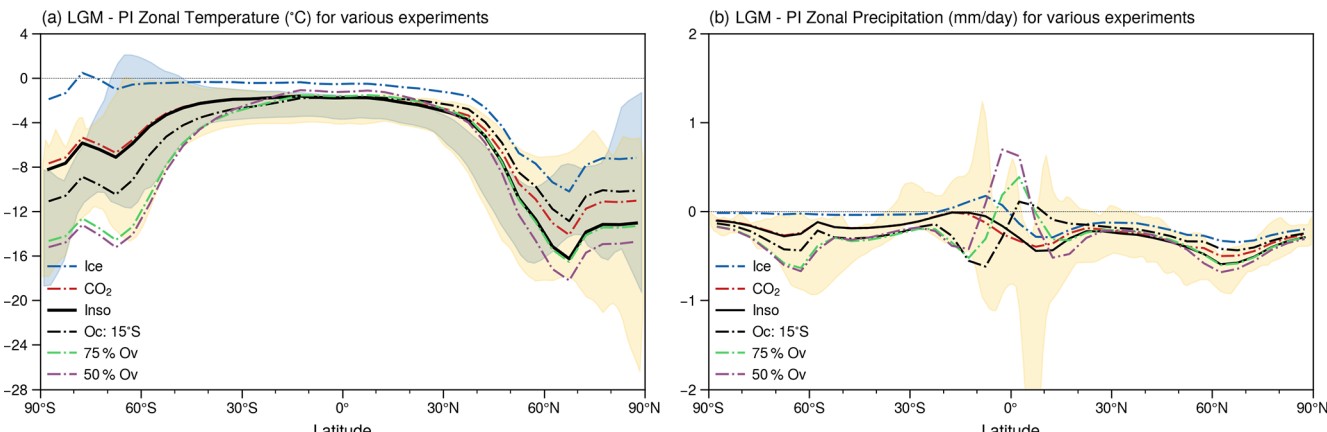

**Figure 10.** The LGM − PI surface air temperatures **(a)** and precipitation rates **(b)** for a series of experiments, which involve progressively adding LGM land ice fractions and zonal-mean land elevations (dash-dotted blue line), LGM $CO_2$ concentrations (dash-dotted red line), LGM insolation (black line), a shift in the mid-point of ocean circulation from 5 to 15° S (dash-dotted black line), a 75 % decrease in ocean overturning (dash-dotted green line) strength, and a 50 % decrease in ocean overturning strength (dash-dotted purple line).

**Table 4.** Difference in zonal- and annual-mean surface air temperatures (°C) between the LGM and the PI. Inso refers to the standard ZEMBA-LGM experiment shown in Fig. 9.

| Model | 90–30° S | 30° S–30° N | 30–90° N | Global |
|---|---|---|---|---|
| ZEMBA | | | | |
| Ice | −0.45 | −0.62 | −5.06 | −1.69 |
| $CO_2$ | −3.53 | −1.93 | −7.69 | −3.77 |
| Inso | −3.69 | −1.95 | −8.83 | −4.11 |
| Oc: 15° S | −5.63 | −1.99 | −6.83 | −4.11 |
| 75 % Ov | −7.74 | −1.90 | −8.88 | −5.11 |
| 50 % Ov | −8.03 | −1.56 | −9.91 | −5.26 |
| Other models and reconstructions | | | | |
| PMIP3 | −3.81 | −2.79 | −9.46 | −4.71 |
| PMIP4 | −4.81 | −2.68 | −8.91 | −4.77 |
| Annan et al. (2022) | −3.57 | −2.63 | −9.02 | −4.46 |

partition of atmospheric heat transport into its dry static and latent components, the model captures the relative contributions of each flux to the total atmospheric heat transport, with dry static fluxes peaking at $\sim 15°$ N/S and latent heat transport peaking at 45° N/S.

Despite these favourable comparisons with state-of-the-art climate models and reanalysis data, there are limitations of ZEMBA in its current state. For example, while the model does reasonably well at capturing the radiative fluxes at the TOA and the surface (Fig. 6), one of the largest discrepancies between ZEMBA and NorESM2 or ERA5 1940–1970 resides in the outgoing longwave radiation at the TOA. As the outgoing longwave flux is overestimated around the tropics, the net radiation received at these latitudes is up to $\sim 25$ W m$^{-2}$ lower than NorESM2 and ERA5 1940–1970 (Fig. 6c). Given the comparatively large surface area in the tropics, the underestimation of these net radiative fluxes re-

sults in a significant reduction in the surplus radiative energy which needs to be transported poleward. Consequently, both the total and the atmospheric heat transport in ZEMBA is much lower (Fig. 7a–b), with atmospheric heat transport peaking at $\sim 3$ PW in each hemisphere, whereas it should be closer to 5 PW (Trenberth and Fasullo, 2017). In addition, one of the challenging aspects of a simple model such as ZEMBA is the underestimation of the winter maximum in snow cover over land (Fig. 5a), resulting in a smaller seasonal amplitude in snow cover. However, we note that the annual-mean rates of snowfall correspond well to NorESM2 and ERA5 1940–1970 over the northern high latitudes (Fig. 3a). Similarly to NH snow cover, the seasonal range of sea ice cover in the SH is reduced when compared to NorESM2 and ERA5 1940–1970 (Fig. 5b) and snowfall rates over Antarctica are underestimated by about 50 % (Fig. 3a).

**Table 5.** Difference in zonal- and annual-mean precipitation rates ($mm\,d^{-1}$) between the LGM and the PI. Inso refers to the standard ZEMBA-LGM experiment shown in Fig. 9.

| Model | 90–30° S | 30° S–30° N | 30–90° N | Global |
|---|---|---|---|---|
| ZEMBA | | | | |
| Ice | −0.03 | −0.07 | −0.22 | −0.10 |
| $CO_2$ | −0.18 | −0.20 | −0.34 | −0.23 |
| Inso | −0.18 | −0.20 | −0.38 | −0.24 |
| Oc: 15° S | −0.29 | −0.20 | −0.29 | −0.25 |
| 75 % Ov | −0.35 | −0.17 | −0.36 | −0.27 |
| 50 % Ov | −0.35 | −0.11 | −0.39 | −0.24 |
| Other models and reconstructions | | | | |
| PMIP3 | −0.25 | −0.25 | −0.42 | −0.29 |
| PMIP4 | −0.30 | −0.30 | −0.43 | −0.33 |

Given their simplicity, the ability of EBMs like ZEMBA to accurately reproduce the latitudinal pattern of surface temperature is impressive but has been well established since the original works by Budyko (1969) and Sellers (1969). In reference to other studies, ZEMBA is most similar to later iterations of EBMs that contain vertical resolution (i.e. an atmospheric and surface layer), the division of the surface into land and ocean, and a seasonal cycle in insolation (Peng et al., 1987; Harvey, 1988; Bintanja, 1997). In comparison to the "present-day" simulation from the EBM developed by Bintanja (1997), which effectively serves as the basis for ZEMBA, most climate variables exhibit qualitative agreement with the pre-industrial output shown in Sect. 3.1, although the global mean surface air temperature reported in Bintanja (1997) is 1.09 °C warmer due to a higher 350 ppm $CO_2$ forcing in their model experiment. Other than the hydrological cycle, the primary contrast between the models lies in their representation of land-based snow cover. In the Bintanja (1997) study, the seasonal cycle in snow cover is parameterized as a function of surface air temperature and appears to be in better agreement with observations than ZEMBA, which explicitly calculates snow coverage over land. Overall, many such features of the Earth's climate are relatively well produced by various EBMs (Peng et al., 1987; Harvey, 1988; Bintanja, 1997), including ZEMBA, in part because they are "tuned" to match the present-day climate. Of greater importance for EBMs than achieving an exact replication of observations and/or GCM output is the investigation of different climate processes and feedbacks, which necessitates appropriate sensitivities to alterations in external or internal forcings.

The responses of ZEMBA to both a doubling of the atmospheric $CO_2$ concentration and a 2 % increase in solar insolation are in qualitative and quantitative agreement with one another (Fig. 8), including polar amplification of warming, which is stronger in the NH than the SH and concentrated in the winter months. The climate sensitivity of ZEMBA is larger than that of the EBM of Bintanja (1997) and similar such studies (Peng et al., 1987). For example, the $S_o + 2\%$ experiment generates much stronger annual-mean warming in the high northern latitudes ($> 14°$) than Bintanja (1997) ($< 5°$). In addition, the ECS of ZEMBA (3.6 °C) is higher than that of the original model of 1.9–2.2 °C (Bintanja, 1997; Stap et al., 2014). However, more recent EBMs have generated global mean and polar-amplified warming in response to $CO_2$ forcing which is similar to or even greater in magnitude than that of ZEMBA (Roe et al., 2015; Södergren et al., 2018; Feldl and Merlis, 2021). Moreover, the ECS of ZEMBA is consistent with the ECS range projected by GCMs that contributed to the Coupled Model Intercomparison Project Phase 5 (CMIP5) and CMIP6 (Flato et al., 2013; Zelinka et al., 2020) and other works (Collins et al., 2013). The strong seasonal asymmetry of surface warming in the polar regions (Fig. 8b–d) is in accordance with both observations (Screen and Simmonds, 2010) and GCM results (Holland and Bitz, 2003). As in Bintanja (1997), the winter maximum in surface air temperature is related to sea ice loss, causing a greater absorption of shortwave radiation, which is released from the ocean surface into the overlying atmosphere during the winter months. For the $2xCO_2$ experiment, the polar amplification in warming – normalized (divided) by the global mean warming – exceeds 4 in the high latitudes of the Arctic and reaches up to 1.5 in the Antarctic (Fig. 8a). While polar amplification in warming that is strongest in the NH is in agreement with both observations and GCM results, warming in the high northern latitudes that reaches 4 times the global mean resides in the upper boundary of estimates from GCM simulations (Holland and Bitz, 2003; Hahn et al., 2021). When averaged from 60 to 90° N, however, the normalized Arctic warming is 3.2, which is still less than the 3.5 estimated over the historical period from 1979 to 2014 according to the observational dataset HadCRUT5 (Hahn et al., 2021). Overall, we see that ZEMBA exhibits a climate sensitivity – in terms of both global mean and polar-amplified

warming – that appears broadly consistent with other EBMs, GCMs and observations.

There are comparatively few EBMs which incorporate a hydrological cycle (Jentsch, 1991; Kukla et al., 2023). Jentsch (1991) developed an EBM, consisting of a vertically averaged atmosphere overlying an ocean layer, to study the influence of the hydrological cycle on climate. Using an optimization procedure, the modelled precipitation and evaporation, amongst other climate variables, compare very well with contemporary observations (Jentsch, 1991). While both ZEMBA and the work of Jentsch (1991) contain a hydrological cycle, ZEMBA includes land cover and a seasonal cycle, which is suitable for investigating the response of climate to variations in the orbital parameters. Kukla et al. (2023) introduced a moist static energy balance model with a hydrological cycle coupled to a carbon cycle model, suited for studying the long-term relationship between the carbon cycle, hydrological cycle and climate. However, the modelled precipitation does not impact the surface albedo, which is instead simplified as a function of surface temperature, and, like the work of Jentsch (1991), it lacks the seasonal insolation cycle necessary to study the climate response to Milanković cycles. Previous EBMs used for studies of glacial–interglacial cycles prescribe either a present-day (Pollard, 1978) or a spatially uniform (Huybers and Tziperman, 2008) distribution of precipitation, which is not perturbed across climates, or instead parameterize precipitation and snowfall as a function of surface air temperatures and ice sheet size (Stap et al., 2014). The inclusion of a hydrological cycle enables precipitation and snowfall to be calculated internally in ZEMBA, which can account for the influence of changes in both local air temperatures and poleward moisture transport for precipitation and snowfall rates. It has been suggested that changes in the Earth's obliquity – by altering the meridional gradient in insolation – have a relatively strong influence on ice sheet volume due to changes in poleward moisture transport (Raymo and Nisancioglu, 2003; Nisancioglu, 2004). The inclusion of a hydrological cycle enables ZEMBA to explore these processes in the context of glacial–interglacial cycles.

It is important that ZEMBA, intended for studies of glacial–interglacial cycles, can simulate climates other than the present-day or pre-industrial periods. When our simulation of the LGM is compared to other reconstructions, the model compares favourably regarding changes in both surface air temperature (Fig. 9a) and precipitation (Fig. 9b). Indeed, ZEMBA captures the polar amplification of cooling in both hemispheres due to positive feedbacks relating to snow cover and sea ice expansion, although this cooling appears somewhat underestimated around Antarctica and slightly overestimated in the northern high latitudes. Moreover, the global mean cooling of $-4.11\,°C$ is similar to that estimated by the ensemble averages from PMIP3 and PMIP4 (Kageyama et al., 2021) and the data assimilation product from Annan et al. (2022). It should be noted that other data assimilation reconstructions suggest much stronger global

cooling of between 6.1 and 6.8 °C (Tierney et al., 2020; Osman et al., 2021), which perhaps signals the importance of including feedbacks relating to clouds, dust and/or vegetation for reproducing the LGM cooling. However, the reconstructions from Tierney et al. (2020) and Osman et al. (2021) are based on a single climate model (CESM1.2), which produces one of the coldest LGM climates in PMIP4, whereas the Annan et al. (2022) reconstruction incorporates the wide range of climates generated across the PMIP ensemble.

Evaluating changes in meridional heat transport during the LGM is made difficult by the large (and often conflicting) range of total, atmospheric and ocean heat transport generated across PMIP3 and PMIP4 at all latitudes (Kageyama et al., 2021). For the ZEMBA-LGM experiment, changes in total heat transport are most intense in the NH (Fig. 9c), in response to large steepening of the Equator-to-pole temperature gradient. The upsurge in total heat transport between 15–45° N is somewhat consistent with PMIP3–PMIP4 (Kageyama et al., 2021), although most of this increase in ZEMBA is generated by stronger atmospheric heat transport (Fig. 9c–d), whereas PMIP3–PMIP4 also shows stronger ocean heat transport at these latitudes. Stronger ocean heat transport in PMIP3–PMIP4 can be attributed to a stronger and sometimes deeper Atlantic Meridional Overturning Circulation (AMOC) (Kageyama et al., 2021), which is inconsistent with proxy-based reconstructions of ocean circulation (Lynch-Stieglitz et al., 2007; Gebbie, 2014; Du et al., 2020) and is associated with stronger surface winds over the northern North Atlantic (Muglia and Schmittner, 2015). For ZEMBA, on the other hand, there is a large decreases in NH ocean heat transport during the LGM, reaching ca. $-0.5\,PW$ at 60° N (Fig. 9c). The prominent reduction in ocean heat transport, not observed in any of the PMIP3–PMIP4 simulations, is associated with the expansion of sea ice and a surface ocean layer that resides near the freezing point of seawater. Consequently, horizontal heat fluxes due to both advection (representing overturning) and diffusion (representing eddies and gyres) drop to zero at latitudes now covered by sea ice, which perhaps highlights the limitation of the simplified ocean model used in ZEMBA, where meridional heat transport is limited to the surface and bottom ocean layers.

The sensitivity experiments performed for the LGM boundary conditions (Fig. 10) recreate the experiment made by Bintanja and Oerlemans (1996) using the original EBM. For ZEMBA, the global mean cooling caused by the combined ice sheet and $CO_2$ forcing is 4.11 °C, which is stronger than the 3.3 °C of cooling noted by Bintanja and Oerlemans (1996) and is in better agreement with LGM reconstructions. The stronger cooling in ZEMBA can be partially attributed to differences in the LGM boundary conditions used in this study, which involves a stronger $CO_2$ reduction and larger LGM ice sheet extents than those in Bintanja and Oerlemans (1996). In our study, the addition of LGM ice sheets generates cooling and drying that are primarily localized to the NH, whereas $CO_2$ lowering generates widespread cool-

ing and drying with stronger global mean cooling than that induced by ice sheet expansion (Tables 4 and 5). Additionally, by shifting the mid-point of ocean circulation from 5 to 15° S, surface air temperatures are in better agreement with the LGM reconstructions, as first noted by Stap et al. (2014). As for the strength of ocean circulation, Bintanja and Oerlemans (1996) found that reducing flow velocities in their ocean model led to temperatures in better agreement with contemporary LGM temperature reconstruction. In this study, however, reductions in ocean overturning instead lead to larger discrepancies with modern LGM constructions. Indeed, it remains uncertain to what degree overturning changed during glacial conditions, with some studies suggesting the mean state of overturning was not "sluggish" across glacial cycles (Bohm et al., 2015) or the LGM (Lynch-Stieglitz et al., 2007). Therefore, for simulations of glacial–interglacial cycles using ZEMBA, it may be important to consider the impact of variations in both the mid-point and the overall strength of ocean circulation.

ZEMBA is intended to be used as a computationally efficient tool for studies of the glacial–interglacial cycles of the Quaternary. The PI and the LGM experiments indicate ZEMBA is able to simulate glacial and interglacial climate states for a given insolation, $CO_2$ and ice sheet extent. As the model includes (1) both hemispheres, (2) a seasonal cycle and (3) a hydrological cycle, it is able to explore mechanisms invoked to explain the dearth of precession cycles in ice volume observed during the Early Pleistocene, such as (1) out-of-phase precession cycles between the hemispheres (Raymo et al., 2006), (2) a counterbalancing between summer insolation intensity and summer duration (Huybers, 2006), or (3) obliquity-induced variations in atmospheric moisture transport (Raymo and Nisancioglu, 2003). In future work, we intend to explore both the equilibrium of ZEMBA and its transient response to changes in the Earth's obliquity and precession, prior to simulations of the glacial–interglacial cycles of the Early Pleistocene via coupling to an ice sheet model.

## 5  Conclusions

In this study, a simple climate model (ZEMBA) is introduced to simulate zonally averaged climate fields including surface temperatures and precipitation. ZEMBA is largely built on the zonally averaged energy balance climate model from Bintanja (1997), comprising an atmospheric layer overlying a surface divided into a land component and a six-layered, zonally averaged ocean transport model. Unlike its predecessor, ZEMBA incorporates a hydrological cycle to estimate snowfall and precipitation with latitude.

Simulations of the pre-industrial period compare favourably with GCMs and reanalysis data, including surface temperatures (and their seasonal cycle), precipitation, surface and TOA radiative fluxes, sea ice, snow cover, and meridional heat transport. However, the underestimation

of the net TOA radiation received in the tropics leads to a reduction in atmospheric heat transport, and there is also an underestimation of the seasonal amplitude in snow cover over land.

The responses of ZEMBA to increases in the atmospheric $CO_2$ concentration or the solar constant are in qualitative agreement with other EBMs, GCMs and observations, such as polar amplification in surface warming, which is strongest over the NH and focused in the winter months. The new additions to ZEMBA appear to increase climate sensitivity in comparison to older EBMs, but its results are still broadly consistent with the global mean and polar-amplified warming projected by climate models of higher complexity.

As the purpose of ZEMBA is for studies of the glacial–interglacial cycles of the Quaternary, it is important that the model can simulate climates other than present-day or pre-industrial climates. A simulation of the LGM indicates ZEMBA is able to capture changes in surface temperature and precipitation in qualitative and quantitative agreement with state-of-the-art climate models and data assimilation products, despite neglecting climate feedbacks relating to dust, vegetation and clouds. In particular, ZEMBA reproduces the polar amplification of cooling in both hemispheres and global mean cooling in accord with reconstructions from more elaborate models.

The overall conclusion from this study is that ZEMBA is suitable for studies of climatic change on large spatial and temporal scales, with a particular emphasis on glacial–interglacial cycles and the response of the climate system to changes in the orbital parameters. In future work, we intend to explore both the equilibrium of ZEMBA and its transient response to changes in the Earth's obliquity and precession, prior to simulations of the glacial–interglacial cycles of the Early Pleistocene via coupling to an ice sheet model.

## Appendix A:  Additional sensitivity experiments

### A1  Sensitivity to prescribed cloud cover

To assess the limitations of prescribing a single cloud cover fraction from a pre-industrial simulation of NorESM2, we repeat the ZEMBA-PI simulation with different choices of zonal-mean cloud cover. In the ZEMBA-PI$_{\text{CESM2}}$ and ZEMBA-PI$_{\text{MRI-ESM2}}$ experiments, we force ZEMBA with pre-industrial cloud cover fractions taken from the Community Earth System Model 2 (CESM2) and the Meteorological Research Institute Earth System Model Version 2.0 (MRI-ESM2), respectively. CESM2 generates larger cloud cover fractions (ranging from 5 % to 25 %) relative to NorESM2 for the pre-industrial period, whereas cloud cover fractions from MRI-ESM2 and NorESM2 correspond more closely (Fig. A1a, c). In the ZEMBA-PI$_{\text{ERA5}}$ and ZEMBA-PI$_{\text{CERES}}$ experiments, ZEMBA is forced with cloud cover fractions taken from the ERA5 atmospheric reanalysis averaged from

1940 to 1970 and from the Clouds and the Earth's Radiant Energy System (CERES) Energy Balanced and Filled product averaged from 2005 to 2015, respectively. Similarly to CESM2, the CERES product contains much larger cloud cover fractions over the tropics and mid-latitudes but noticeably less cloud cover in the polar latitudes (Fig. A1a, c). On the other hand, the ERA5 dataset contains similar cloud cover to NorESM2 at the lower latitudes but has higher cloud cover in the polar regions (Fig. A1a, c).

Figure A1b shows the surface temperatures generated by ZEMBA when forced by these different cloud cover fractions, and Fig. A1d shows these anomalies relative to the standard PI simulation of ZEMBA (using NorESM2 PI cloud cover). When ZEMBA is forced by MRI-ESM2 and ERA5 cloud cover, which correspond closely to NorESM2 over most of the tropics and mid-latitudes, the differences in zonal-mean temperature are quite small. Changes in global mean temperature do not exceed 0.25 °C between these three PI simulations. However, when forced by CESM2 or CERES cloud cover fractions, which are generally much higher than NorESM2, there is a strong cooling effect. The decrease in temperature is unsurprising given that the net effect of more cloud cover is to cool the Earth for the radiation parameterization used in ZEMBA from Bintanja (1996). However, the global mean cooling for CERES (2.4 °C) is larger than that of CESM2 (1.67 °C), despite CESM2 having a consistently larger cloud cover. In particular, using CERES cloud cover generates much stronger cooling at high latitudes, in the region where CERES has very low cloud cover and where CESM2 has very high cloud cover. This suggests that, while the net global effect of clouds is to cool the Earth, the warming effect of clouds (via longwave radiation) outweighs the cooling effect (via shortwave radiation) in the polar regions. Indeed, ERA5 also generates warmer temperatures in the polar regions, where cloud cover fractions are higher than for NorESM2. Overall, we see that the choice of cloud cover fractions can have a strong impact on surface air temperature for the pre-industrial period. While the differences are small compared to MRI-ESM2 and ERA5 cloud cover fractions, they can become substantial for CESM2 and CERES. The choice of a different cloud cover fraction in ZEMBA would require a retuning of other model parameters to ensure the model simulates surface air temperatures with reasonable accuracy for the pre-industrial period.

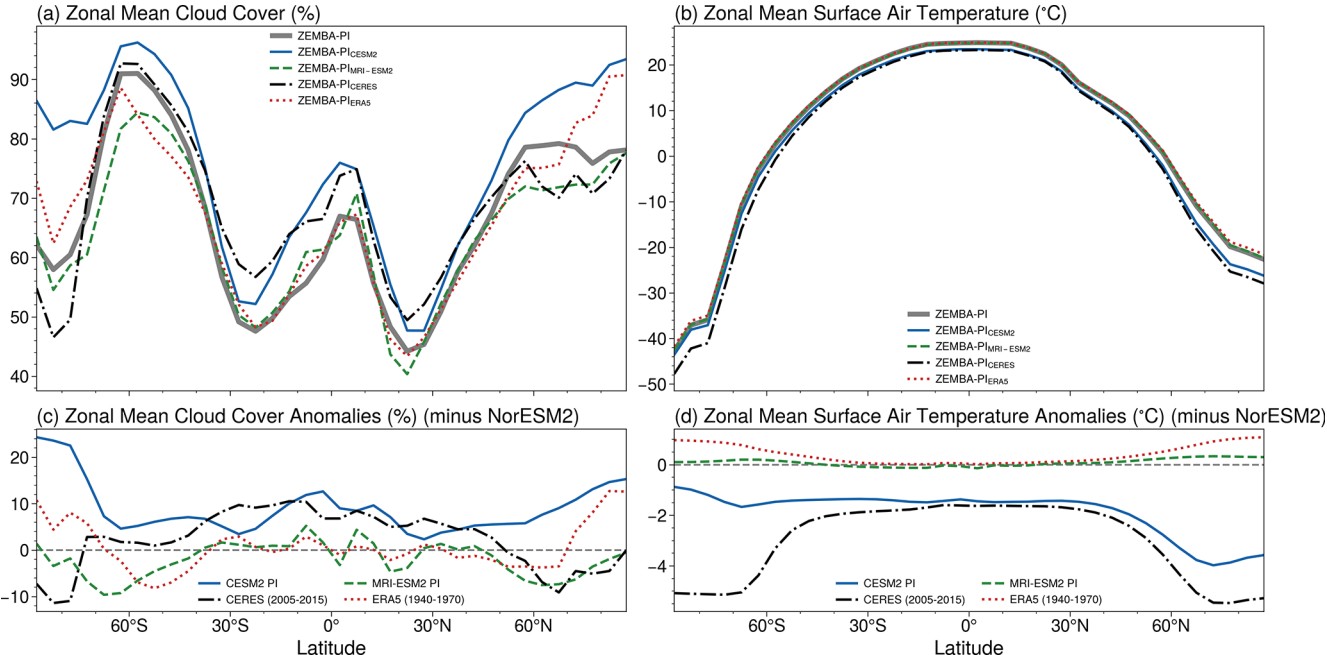

**Figure A1. (a)** Different values of zonal-mean cloud cover including NorESM2 (ZEMBA-PI in grey), CESM2 (ZEMBA-PI$_{CESM2}$ in blue), MRI-ESM2 (ZEMBA-PI$_{MRI-ESM2}$ in green), CERES 2005–2015 (ZEMBA-PI$_{CERES}$ in black) and ERA5 1940–1970 (ZEMBA-PI$_{ERA5}$ in red); **(b)** the zonal-mean surface air temperature simulated by ZEMBA in response to these different cloud cover fractions; **(c)** the differences in zonal-mean cloud cover relative to NorESM2 PI cloud cover; and **(d)** the differences in zonal-mean surface air temperature (caused by using different cloud cover fractions) relative to the standard ZEMBA-PI simulation. Pre-industrial cloud cover from CERES and MRI-ESM2 is taken from the Earth System Grid Federation at https://esgf-node.llnl.gov/search/cmip6/ (last access: 25 October 2024). ERA5 cloud cover is taken from Hersbach et al. (2023) (https://doi.org/10.24381/cds.f17050d7, last access: 25 October 2024), and CERES is taken from https://ceres.larc.nasa.gov/data/ (last accessed: 25 October 2024).

## A2   Sensitivity to key model parameters

We examine the sensitivity of ZEMBA to internal model parameters by replicating selected experiments from Bintanja (1997). In addition to the $CO_2$ (ZEMBA-$2 \times CO_2$) and solar-constant (ZEMBA-$S_o + 2\%$) experiments described earlier, we perform simulations with perturbations in cloud cover amount ($+20\%$), the cloud optical depth ($\tau + 1.4$), the diffusion coefficients for both horizontal ($D_o \times 2$) and vertical ($D_z \times 2$) ocean heat transport, the turbulent heat flux coefficient ($\kappa \times 2$), and sea ice thickness ($d_{si} \times 2$). We also perform a new experiment with a perturbation in a Hadley cell constant ($\lambda + 0.6$). Figure A2 shows the changes in global mean surface air temperature, global mean ocean temperature and the average Equator-to-pole temperature gradient driven by these large perturbations in model parameters.

The response of ZEMBA to perturbations in these model parameters is qualitatively similar to the responses reported by Bintanja (1997), though ZEMBA shows heightened sensitivity, as identified previously in the ZEMBA-$2 \times CO_2$ and ZEMBA-$S_o + 2\%$ experiments. This increased climate sensitivity in ZEMBA may be attributed to changes such as the parameterization of land surface albedo and the use of atmospheric heat transport set proportional to gradients in moist static energy, rather than temperature, which can enhance polar-amplified warming. Both the $2\%$ increase in the solar constant and the $CO_2$ doubling yield similar outcomes: global mean and polar-amplified warming leading to a reduced Equator-to-pole temperature gradient.

Perturbations in the cloud cover parameters cause the most significant changes in surface temperature. The net effect of increased cloud cover is to drive global mean cooling. Moreover, an increase in cloud optical depth ($\tau$), which increases the cloud albedo, produces an even stronger cooling effect than changes in total cloud amount. As noted by Bintanja (1997), the stronger sensitivity to an increased cloud optical depth is due to its sole effect of enhancing the reflection of incoming shortwave radiation at the TOA. In contrast, when

cloud cover is increased, the enhanced shortwave reflection is partially offset by decreased outgoing longwave radiation, which moderates the overall cooling effect.

Adjustments to the ocean heat transport coefficients, turbulent heat flux coefficients or sea ice thickness have more modest impacts on global mean temperature compared to cloud cover. Doubling the eddy and gyre diffusion coefficient ($D_o$) marginally raises polar temperatures without significantly affecting the global mean. Similarly, doubling the vertical diffusion coefficient ($D_z$) has a negligible effect on air temperature but markedly affects mean ocean temperature. It should be noted, however, that a large proportion of ocean heat transport is carried out by the prescribed ocean overturning, which is unaffected by these diffusion coefficients. Increasing $\kappa$ enhances the surface-to-atmosphere heat fluxes, which leads to surface cooling and thereby sea ice and snow expansion and ultimately results in global mean cooling. Increasing $d_{si}$ results in a slight global temperature decrease by reducing the seasonal variability in sea ice, thereby increasing its extent during summer months (Bintanja, 1997).

Finally, to evaluate sensitivity related to the Hadley cell parameterization, we modify $\lambda$ – representing the fractional difference between the upper branch's uniform moist static energy and the surface moist static energy at the Equator ($\overline{m}_{eq}$). In the original formulation from Siler et al. (2018), $\lambda = 1.06$, which we adjusted to 1.03 for an improved simulation of PI precipitation. In this sensitivity experiment, we increase $\lambda$ to 1.09. While changing $\lambda$ affects zonal precipitation in the tropics, the total atmospheric energy transport remains governed by meridional gradients in moist static energy, leaving global mean temperature largely unaffected. Additionally, assumptions involving the $\Omega$ weighting function, which determines the Hadley cell's dominance in heat transport, were explored by Siler et al. (2018), who noted that varying representations of $\Omega$ did not alter the primary climate response.

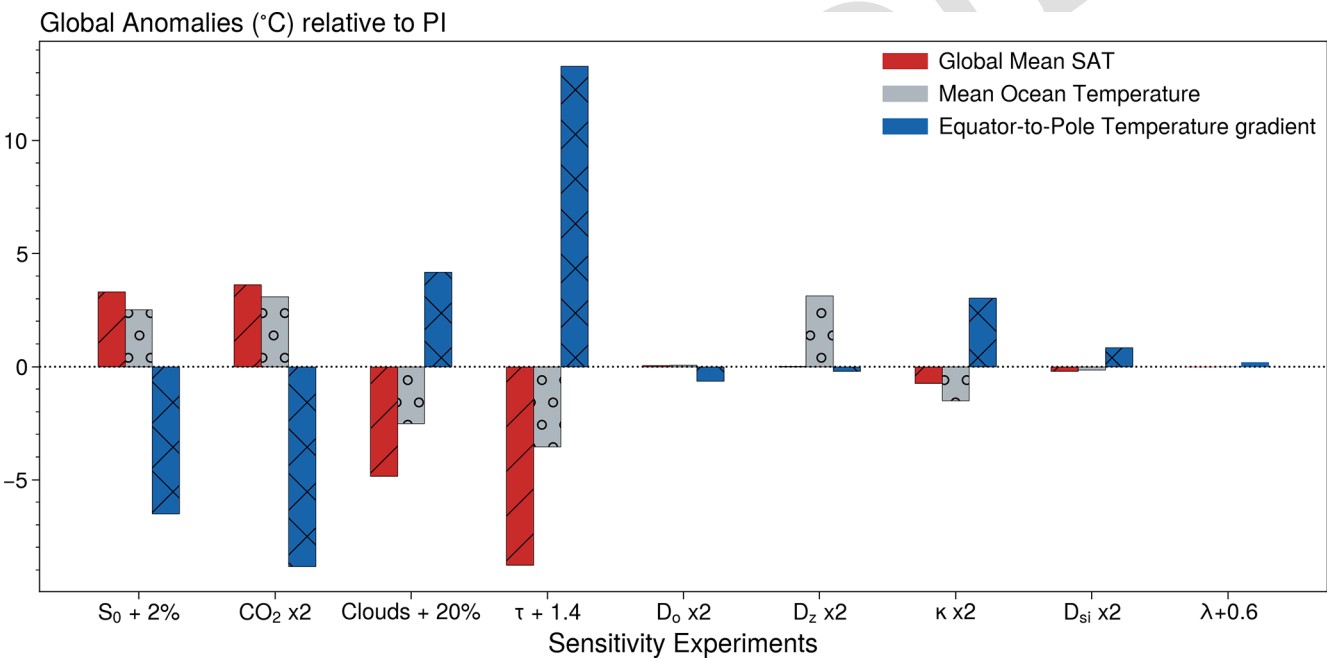

**Figure A2.** Anomalies in global mean surface air temperature, global mean ocean temperature, and the air temperature difference between the equatorial (0–10°) and polar regions (80–90°) for changes in the solar constant ($S_o$), atmospheric $CO_2$ level ($CO_2$), cloud amount, cloud optical depth ($\tau$), ocean diffusion coefficient for horizontal eddy and gyre heat transport ($D_o$), ocean diffusion coefficient for vertical heat transport ($D_z$), coefficient for turbulent heat fluxes ($\kappa$), sea ice thickness ($d_{si}$) and a Hadley cell parameter ($\lambda$).

## Appendix B: List of model parameters

In Table B1 we provide definitions for all the model parameters listed in Sect. 2.

**Table B1.** All model parameters listed in the atmospheric, land and ocean components of the model.

| Parameter | Units | Value | Description |
|---|---|---|---|
| **General** | | | |
| $\phi$ | degrees | – | Latitude |
| $R_e$ | m | $6.371 \times 10^6$ | Earth's radius |
| i | – | – | Index for land or ocean grid cell |
| **Atmospheric, land and ocean properties** | | | |
| $T_{a(i)}$ | K | – | Near-surface temperature of the atmospheric layer (over land or ocean) |
| $T_{a(i)'}$ | K | – | Near-surface temperature of the atmospheric layer (over land or ocean) corrected for zonal-mean elevation |
| $\overline{T_a}$ | K | – | Near-surface temperature of the atmospheric layer (zonal average) |
| $C_a$ | $J\,kg^{-1}\,K^{-1}$ | 1004 | Specific heat capacity of air |
| $H_a$ | m | 8194 | Height/thickness of the atmospheric layer |
| $\rho_a$ | $kg\,m^{-3}$ | 1.25 | Density of air |
| $T_o$ | K | – | Ocean temperature |
| $C_o$ | $J\,kg^{-1}\,K^{-1}$ | 3850 | Specific heat capacity of seawater |
| $H_o$ | m | 100, 316.6, 543.5, 775.8, 1012.3, 1251.8 | Thickness of each ocean layer in descending order |
| $\rho_o$ | $kg\,m^{-3}$ | 1025 | Density of seawater |
| $T_l$ | K | – | Land surface temperature |
| $T_l'$ | K | – | Land surface temperature corrected for land elevation |
| $C_l$ | $J\,kg^{-1}\,K^{-1}$ | 1480 | Specific heat capacity of land |
| $H_l$ | m | 2.2 | Thickness of layer |
| $\rho_l$ | $kg\,m^{-3}$ | 2000 | Density of ground layer |
| **Radiative fluxes** | | | |
| $S_{(i)}$ | $W\,m^{-2}$ | – | Absorbed shortwave radiation at the surface (over land or ocean) |
| $S_{BOA(i)}^{\downarrow}$ | $W\,m^{-2}$ | – | Incoming shortwave radiation at the surface (over land or ocean) |
| $S_{BOA(i)}^{\uparrow}$ | $W\,m^{-2}$ | – | Outgoing shortwave radiation at the surface (over land or ocean) |
| $S_{a(i)}$ | $W\,m^{-2}$ | – | Absorbed shortwave radiation in the atmospheric layer (over land or ocean) |
| $S_{TOA(i)}^{\downarrow}$ | $W\,m^{-2}$ | – | Incoming shortwave radiation at the top of the atmosphere (over land or ocean) |
| $S_{TOA(i)}^{\uparrow}$ | $W\,m^{-2}$ | – | Outgoing shortwave radiation at the top of the atmosphere (over land or ocean) |
| $I_{(i)}$ | $W\,m^{-2}$ | – | Absorbed longwave radiation at the surface (over land or ocean) |
| $I_{BOA(i)}^{\downarrow}$ | $W\,m^{-2}$ | – | Incoming longwave radiation at the surface |
| $I_{BOA(i)}^{\uparrow}$ | $W\,m^{-2}$ | – | Outgoing longwave radiation at the surface |
| $I_{a(i)}$ | $W\,m^{-2}$ | – | Absorbed longwave radiation in the atmospheric layer (over land or ocean) |
| $I_{TOA(i)}^{\uparrow}$ | $W\,m^{-2}$ | – | Outgoing longwave radiation at the top of the atmosphere |
| $\tau$ | – | 3.0 | Cloud optical depth |
| $\theta_z$ | – | – | Zenith angle of incoming shortwave radiation at the top of the atmosphere |

| Parameter | Units | Value | Description |
|-----------|-------|------:|-------------|
| **Turbulent heat fluxes** | | | |
| $K_{(i)}$ | $\mathrm{W\,m^{-2}}$ | – | Sensible heat flux from the surface (over land or ocean) |
| $E_{(i)}$ | $\mathrm{kg\,m^{-2}\,s^{-1}}$ | – | Evaporation from the surface (over land or ocean) |
| $\kappa_\mathrm{l}$ | $\mathrm{m\,s^{-1}}$ | 0.01 | Turbulent heat flux coefficient over land |
| $\kappa_\mathrm{o}$ | $\mathrm{m\,s^{-1}}$ | 0.006 | Turbulent heat flux coefficient over ocean |
| $W_\mathrm{l}$ | – | 0.7 | Surface water availability over land |
| $W_\mathrm{o}$ | – | 1.0 | Surface water availability over ocean |
| **Hydrological cycle** | | | |
| $Q_{\mathrm{a}(i)}$ | $\mathrm{kg\,kg^{-1}}$ | – | Specific humidity of the atmospheric layer (over land or ocean) |
| $\overline{Q_\mathrm{a}}$ | $\mathrm{kg\,kg^{-1}}$ | – | Specific humidity of the atmospheric layer (zonal average) |
| $L_\mathrm{v}$ | $\mathrm{J\,kg^{-1}}$ | $2.5 \times 10^6$ | Latent heat of vaporization |
| $P_{\mathrm{r}(i)}$ | $\mathrm{kg\,m^{-2}\,s^{-1}}$ | – | Precipitation flux (over land or ocean) |
| $L_\mathrm{f}$ | $\mathrm{J\,kg^{-1}}$ | $3.34 \times 10^5$ | Latent heat of fusion |
| $P_{\mathrm{s}(i)}$ | $\mathrm{kg\,m^{-2}\,s^{-1}}$ | – | Snowfall flux (over land or ocean) |
| $r_\mathrm{max}$ | – | 80 | Maximum relative humidity |
| $r$ | – | – | Relative humidity of the atmospheric layer |
| $f_\mathrm{sf}$ | – | – | Geographic fraction of precipitation that falls as snow |
| **Surface albedo** | | | |
| $\alpha_\mathrm{s}$ | – | – | Snow albedo |
| $\alpha_\mathrm{cs}$ | – | 0.8 | Maximum "cold" snow albedo |
| $\alpha_\mathrm{ws}$ | – | 0.4 | Minimum "warm" snow albedo |
| $\alpha_\mathrm{bg}$ | – | 0.15 | Bare-ground albedo |
| $\alpha_\mathrm{bi}$ | – | 0.8 | Land ice albedo |
| $\alpha_\mathrm{g}$ | – | – | Average albedo over bare ground (including snow cover) |
| $\alpha_\mathrm{i}$ | – | – | Average albedo over ice (including snow cover) |
| $\alpha_\mathrm{l}$ | – | – | Average albedo over land |
| $f_\mathrm{i}$ | – | – | Fractional area of land covered by ice |
| $\alpha_\mathrm{op}$ | – | – | Albedo of open ocean |
| $\alpha_\mathrm{si}$ | – | 0.7 | Albedo of sea ice |
| $\alpha_\mathrm{o}$ | – | – | Albedo of ocean |
| $f_\mathrm{si}$ | – | – | Fractional area of ocean covered by sea ice |
| **Snow cover and sea ice** | | | |
| $f_\mathrm{sc}$ | – | – | Fractional area of land covered by snow |
| $d_\mathrm{sc}$ | m | – | Average thickness of the snowpack |
| $\dot{M}$ | m | – | Melt of snowpack |
| $T_K$ | K | 273.15 | Melting point of snow |

| Parameter | Units | Value | Description |
|---|---|---|---|
| $T_{\mathrm{fo}}$ | K | 271.15 | Freezing point of seawater |
| $Q_{\mathrm{si}}$ | W | – | Heat available for growth or melting of sea ice |
| $A_{\mathrm{o}}$ | $\mathrm{m}^2$ | – | Surface area of the ocean |
| $V_{\mathrm{si}}$ | $\mathrm{m}^3$ | – | Sea ice volume |
| $A_{\mathrm{si}}$ | $\mathrm{m}^2$ | – | Sea ice area |
| $d_{\mathrm{si}}$ | m | 2 | Thickness of sea ice |
| $\rho_{\mathrm{ice}}$ | $\mathrm{kg\,m^{-3}}$ | 917 | Density of ice |
| Atmospheric transport | | | |
| $D_{\mathrm{a}}$ | $\mathrm{m\,s^{-1}}$ | $0.7 \times 10^6$ (SH); $0.84 \times 10^6$ (NH) | Diffusion coefficient for total atmospheric heat transport |
| $F_{\mathrm{total}}$ | W | – | Northward flux of moist static energy |
| $F_{\mathrm{HC}}$ | W | – | Northward flux of moist static energy carried out by the Hadley cell |
| $F_{\mathrm{eddy}}$ | W | – | Northward flux of moist static energy carried out by eddies |
| $F_T$ | W | – | Northward flux of dry static energy |
| $F_{T\_\mathrm{HC}}$ | W | – | Northward flux of dry static energy carried out by the Hadley cell |
| $F_{T\_\mathrm{eddy}}$ | W | – | Northward flux of dry static energy carried out by eddies |
| $F_Q$ | W | – | Northward flux of latent energy (moisture) |
| $F_{Q\_\mathrm{HC}}$ | W | – | Northward flux of latent energy carried out by the Hadley cell |
| $F_{Q\_\mathrm{eddy}}$ | W | – | Northward flux of latent energy carried out by eddies |
| $\Omega$ | – | – | Fractional proportion of $F_{\mathrm{total}}$ carried out by the Hadley cell |
| $\overline{m}$ | $\mathrm{J\,kg^{-1}}$ | – | The zonal-average moist static energy of the atmospheric layer |
| $\overline{m}_{\mathrm{eq}}$ | $\mathrm{J\,kg^{-1}}$ | – | The zonal-average moist static energy of the atmospheric layer at the Equator |
| $g$ | $\mathrm{J\,kg^{-1}}$ | – | Difference in moist static energy between the upper and lower branch of the Hadley cell |
| $\lambda$ | – | – | Fractional increase in moist static energy in the upper branch of the Hadley cell relative to $\overline{m}_{\mathrm{eq}}$ |
| $\psi$ | $\mathrm{kg\,s^{-1}}$ | – | Mass transport in either the upper or the lower branch of the Hadley cell |
| Ocean heat transport | | | |
| $z$ | – | – | Vertical coordinate for ocean transport model |
| $D_{\mathrm{o}}$ | $\mathrm{m\,yr^{-1}}$ | $5 \times 10^{10}$ | Diffusion coefficient for horizontal heat transport by eddies and gyres |
| $D_{\mathrm{i}}$ | $\mathrm{m\,yr^{-1}}$ | $1.5 \times 10^{10}$ | Diffusion coefficient for horizontal heat transport in ocean interior |
| $D_z$ | $\mathrm{m\,yr^{-1}}$ | $5 \times 10^3$ | Diffusion coefficient for vertical heat transport |
| $F_{\mathrm{ov}}$ | W | – | Northward flux of ocean heat transport in the top and bottom layer driven by (advective) overturning |
| $F_{\mathrm{eg}}$ | W | – | Northward flux of ocean heat transport in the top layer driven by (diffusive) eddies and gyres |
| $F_{\mathrm{i}}$ | W | – | Northward flux of ocean heat transport in the ocean interior driven by diffusion |
| $f$ | – | – | Fractional width of ocean basin |
| $w$ | $\mathrm{m\,yr^{-1}}$ | – | Prescribed vertical ocean velocities |
| $u$ | $\mathrm{m\,yr^{-1}}$ | – | Prescribed horizontal ocean velocities (in top and bottom ocean layer) |

*Code and data availability.* Source code is maintained on GitHub at https://github.com/daniel-francis-james-gunning/zemba (last access: 8 May 2024) with the exact version used in this study (including scripts for creating all figures) archived on Zenodo at https://doi.org/10.5281/zenodo.11155259 (Gunning, 2024).

*Author contributions.* DFJG designed the model and ran the experiments. DFJG, KHN, EC and RSWvdW analysed the results. DFJG drafted the paper, with input from all co-authors.

*Competing interests.* The contact author has declared that none of the authors has any competing interests.

ther geographical representation in this paper. While Copernicus Publications makes every effort to include appropriate place names, the final responsibility lies with the authors.

*Acknowledgements.* This publication was generated in the frame of DEEPICE project. The project has received funding from the European Union's Horizon 2020 research and innovation programme under Marie Skłodowska-Curie grant agreement no. 955750. Emilie Capron acknowledges financial support from the French National Research Agency under the "Programme d'Investissements d'Avenir" (ANR-19-MPGA-0001). Kerim H. Nisancioglu acknowledges financial support from the Climate Narratives project 324520 funded by the Research Council of Norway.

*Financial support.* This research has been supported by the European Union's Horizon 2020 research and innovation programme under Marie Skłodowska-Curie grant agreement no. 955750. TS4

*Review statement.* This paper was edited by Olivier Marti and reviewed by two anonymous referees.

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

**Remarks from the language copy-editor**

CE1    Please confirm the adjusted values.

**Remarks from the typesetter**

TS1    Please check all affiliation and affiliation codes carefully and confirm if they are correct.

TS2    Please give an explanation of why the number in the equation needs to be changed. We have to ask the handling editor for approval. Thanks.

TS3    Please give an explanation of why the symbol in the equation needs to be changed. We have to ask the handling editor for approval. Thanks.

TS4    Please confirm both Acknowledgements and Financial support sections and make sure that all funders and grant numbers are mentioned in the Financial support section. Thank you.

TS5    DOIs do not need dates of last access.

TS6    Please provide date of last access.