# Peer review of "ZEMBA v1.0: An energy and moisture balance climate model to investigate Quaternary climate"

_EGUsphere, 2024_

## Author Comment (AC1)

The authors wish to thank the reviewer for their constructive and insightful comments which have greatly improved the manuscript. We address the comments as detailed in the following text and in the revised manuscript. The reviewers' comments are in bold underline and the normal text is our responses.

**"- equation (1): Ls should be replaced by Lf (Latent heat of fusion, described later on)."**:

This mistake has been amended in the revised manuscript.

**"- the description of the Hadley cell circulation is a bit difficult to follow with the use of many intermediate notations for heat fluxes. Maybe a final summary of FT and FQ as functions of Ftotal , FT_eddy and FQ_eddy would be useful (equations 18a, 19a)."**

We have redefined Eq. 18 as FT = FT_eddy + FT_HC = FT_eddy [long form defined in Eq. 14] + FT_HC [long form defined in Eq. 16].. and same for FQ in Eq. 19...

**"- there is apparently no vegetation on land and the surface albedo is controlled only by snow... I am wondering to what extent this explains some biaises of the model, or if this is negligible when compared to other factors (cloud cover, fresh snow versus ice albedo, ...). More importantly, it is not clearly explained in the manuscript how land evaporation is computed: there is a "surface water availability" parameter in equation (7), but no information is given on what it actually means. A bit more discussion on land cover (or lack of) would be appreciated."**

As pointed out by the reviewer, we assume a fixed uniform surface albedo over bare ground (of 0.15), unaffected by different vegetation types. In the EBM of Bintanja (1997) land is divided into three surface types (grass, forest and permanent land ice) – all fixed to present-day distributions. The simplification in ZEMBA, specifying one surface albedo for bare land, could lead to discrepancies in the surface albedo for simulations of the pre-industrial climate, particularly over regions with dense forest cover. Including a present-day vegetation distribution and albedo would improve the PI simulation. However, vegetation feedbacks would not be included unless dynamic vegetation cover is included. Given that our study focuses on the feedbacks due to snow, sea ice and ice cover changes we have chosen to exclude changes in vegetation. In future work, it would be interesting to study the impact of dynamic vegetation changes over the Quaternary. In the revised manuscript [**line 265**], we have added the following text to the "Land Surface Albedo" section of the Methods:

*"The assumption of a uniform $\alpha_{bg}$ albedo overlooks the important influence that different vegetation types have on land albedo. In contrast, Bintanja et al., (1997) divides "ice-free" land into present-day distributions of grass and forest cover, though these proportions are held constant over time. Thus, in both approaches, these potentially significant vegetation feedbacks are excluded from Quaternary climate simulations. While including present-day vegetation distribution could improve pre-industrial simulations of ZEMBA, we see limited added value in doing so for studies of orbitally-driven climate change. Nonetheless, we recognize that these simplifications in land albedo may affect the strength of albedo feedbacks over land, which could be explored in future applications of the model."*

As for evaporation, we have included information on the surface water availability parameter which is 1.0 over ocean, and 0.7 over land, reflecting reduced water availability over land (as in Bintanja, 1997). The following text has been added to the Turbulent Heat Fluxes section of the Methods [**line 162**]:

*"Following Bintanja et al., 1997, W is set to 0.7 and 1.0 over land and ocean, respectively, to reflect reduced water availability over land."*

**"- Table 1 lists only a small set of the parameters used in the model. It would be useful to have a more extended list...Examples:"**

We have included a table of all parameters listed in the text in the appendix.

**"line 177: Ta' is Ta corrected with a lapse rate of -6.5K/km. This information is useless if we don't know the height at which Ta' is evaluated."**

At the beginning of the results section, we state "zonal mean elevations over land taken from ICE-6G_C" dataset for the LGM and present-day conditions (also outlined in Table 2). In the revised Methods section, when describing the Hydrological Cycle, we have added a note to make this clear [**line 182**]:

*"In the current version of the model, this zonal-mean elevation is prescribed (see Section 2.4 and Table 2). In the future, we intend to make the zonal-mean elevation dependent on a coupled ice sheet model."*

**"Line 309: constant sea ice thickness..."**

We have clarified that sea ice thickness is set to 2 m (which has been given a symbol – $d_{si}$), both in the text and the table of important model parameters. In the revised text, the following has been added [**line 326**]:

*"Sea ice volume is then converted into sea ice areal extent by assuming a constant sea ice thickness ($d_{si}$) which is set to 2 m".*

**- cloud cover is taken from NorESM and is shown of Fig.1 along some other fixed parameters (ocean circulation, Hadley cells). There is some discussion in the paper of the impact of ocean circulation... but little discussion on the Hadley cell parameters, and none on the clouds. This should certainly be addressed in a revised version of the paper.**

Anonymous Referee #2 has also commented on the need for more discussion on the sensitivity of ZEMBA to cloud cover and other key parameters. We have included a new set of model simulations, to addresses these concerns (as detailed in the reply to reviewer #2). To summarise briefly, we perform an additional set of sensitivity experiments attached in the appendix (Fig. 1), which replicates the large perturbations to model parameters performed by Bintanja (1997). Overall, ZEMBA shows a heightened sensitivity to cloud parameters, and especially the globally averaged cloud optical depth parameter. As for the Hadley cell parameterisation, we modify λ-representing the fractional difference between the upper branch's uniform moist static energy and the surface moist static energy at the equator. While changing λ affects zonal precipitation in the tropics, the total atmospheric energy transport remains governed by meridional gradients

in moist static energy, leaving global mean temperature largely unaffected. For more details, see the appendix of the revised manuscript.

[Figure]

*Figure 1: Anomalies in global mean surface air temperature, global mean ocean temperature and the air temperature difference between the equatorial (0°-10°) and polar regions (80°-90°) for changes in the solar constant (SO), atmospheric CO2 level (CO2), cloud amount, cloud optical depth (τ), ocean diffusion coefficient for horizontal eddy/gyre heat transport (Do), ocean diffusion coefficient for vertical heat transport (Dz), coefficient for turbulent heat fluxes (κ), sea ice thickness (dsi) and a Hadley cell parameter (λ).*

**- line 174: fsf is the fraction of precipitation that falls as snow. Usually, this is understood as a statistical, or possibly as a time fraction. But later on (line 233) this is used as a geographic fraction, which might be something quite different… Is this really justifiable?**

The expression for $F_{sf}$ is taken from Harvey et al., (1988) as the fractional area of the grid box over which (prescribed) precipitation falls as snow, based on meteorological station data from 30N to 90N. In Bintanja et al. (1997), they take this same expression to represent the fractional area of snow present on land for each grid box as a function of surface air temperature. In our case, we use this expression as in Harvey et al. (1988), but instead representing the fractional area of precipitation that falls as snow based on simulated (and not prescribed) precipitation. Given the simplicity of the model, this rate of precipitation is assumed to be uniform for each grid box, thus the geographic fraction of snowfall is equal to the total fraction of precipitation that falls as snow (as presented in the Hydrological cycle). We acknowledge that this is a simplification in the model and have made our approach clearer by modifying the text in the Hydrological Cycle subsection of the *Methods* [**line 183**]*:*

*"The expression for $f_{sf}$ is taken from Harvey (1988) as the fractional area of a grid box over which precipitation falls as snow, based on meteorological station data. Therefore, rather than assuming a uniform distribution of snowfall across each grid box, this parameterization allows for only a portion of the land or ocean surface to be snow-covered. As precipitation is assumed to fall uniformly over each grid box, however, this geographic fraction also represents the overall proportion of precipitation that is converted into snow."*

**- the results in terms of snow cover (Figure5, lines 385) are a bit disappointing. Concerning a future application of this model to the question of ice ages (like a coupling with an ice-sheet model), this might be a severe limitation. Some discussion on this point would be useful... Can really the model be used for such a purpose?**

We acknowledge that the representation of seasonal snow cover in the simple EMBA is challenging. Indeed, for the coupling of ZEMBA to an ice sheet model this needs to be further investigated and improved. However, we find the annual-mean snowfall rates over the NH (Fig. 3) to correspond well with NorESM2 and ERA5 output given the simplicity of the model. In the revised Discussion section (in the paragraph concerning model limitations), we have rewritten the discussion of seasonal snow cover as follows [**line 543**]:

*"In addition, one of the challenging aspects of a simple model such as ZEMBA is the underestimation of the winter maximum in snow cover over land (Fig. 5a), resulting in a smaller seasonal amplitude in snow cover. However, we note that the annual mean rates of snowfall correspond well with NorESM2 and ERA5 1940-1970 over the northern high latitudes (Fig. 3a)."*

**.- almost all model outputs are compared to GCM outputs, except for the meridional heat fluxes on Figure 9c and 9d. This is a bit surprising and, if possible, it should be corrected.**

We opted not to include ERA5 meridional heat transports as the time required to process the large amount of data for comparing to ZEMBA was beyond the scope of this paper. However, we note that the NorESM2 heat transport corresponds nicely to ERA-Interim reanalysis estimates of heat transport averaged from 2000 to 2014 (Trenberth and Fasullo, 2017). In the revised manuscript (in the Pre-Industrial Simulation subsection of the Results section) we have included the following text [**line 426**]:

*"The simulated northward heat transport via the atmosphere and ocean is depicted in Figure 7 in reference to NorESM2. We note that NorESM2 heat transport values replicate those estimated from 2000 to 2014 using ERA-interm reanalysis (Trenberth and Fasullo, 2017), including total heat transport exceeding 5.5 PW in each hemisphere and ocean heat transport peaking around 2 PW at 15N."*

---

## Author Comment (AC2)

The authors wish to thank the reviewers for their constructive and insightful comments which have greatly improved the manuscript. We address the comments as detailed in the following text and in the revised manuscript. The reviewers' comments are in bold underline and the normal text is our responses.

**The model does have some limitations related to its parameterizations, including using a single cloud cover fraction from pre-industrial simulations. One test of the importance of this feature could be testing a range of possible cloud cover fractions (from 2xCO2 experiments, for instance) in ZEMBA. Brief mention of which parameters in Table 1 have (1) high uncertainty and (2) significant impact on the modeled climate would be helpful, as would references for the chosen parameter values. How are the parameters tuned, and are their values within the accepted range of uncertainty? For example, the chosen diffusion coefficient Da is slightly higher than estimates from GCMs (~1.05 x 10^6, e.g., Ge et al., 2023; "The sensitivity of climate and climate change to the efficiency of atmospheric heat transport").**

Following the suggestions by both reviewers, we have assessed the limitations of prescribing a single cloud cover fraction from a pre-industrial simulation of the Norwegian Earth System Model Version 2 (NorESM2). In the revised manuscript, we have included a section of the appendix (A1) where we perform the same pre-industrial simulation of ZEMBA but with cloud cover taken from the Community Earth System Model 2 (CESM2), the Meteorological Research Institute Earth System Model Version 2.0 (MRI-ESM2.0), ERA5 atmospheric reanalysis averaged from 1940 to 1970 and from the Clouds and Earth's Radiant Energy Systems Energy Balanced and Filled (CERES) product averaged from 2005 to 2015. Figure 1 (presented in the appendix of the revised manuscript) shows the different cloud cover fractions used (Fig. 1a,c) and their influence on surface air temperature in ZEMBA (Fig. 1b,d). To summarise, when ZEMBA is forced by MRI-ESM2 and ERA5 cloud cover, which correspond closely to NorESM2 over most of the tropics and mid-latitudes, the differences in zonal mean temperature are quite small (less than 0.25℃). The CESM2 and CERES products, however, contain much higher percentages of cloud cover (relative to NorESM2) over the tropics and mid-latitudes, which generates a strong cooling effect in ZEMBA. Using CESM2 and CERES produces a global mean cooling- relative to standard PI simulation of ZEMBA- of 1.67℃ and 2.4℃, respectively. Overall, the choice of cloud cover fractions can have a strong impact on surface air temperature for the pre-industrial. A different choice of cloud cover fraction in ZEMBA would require a retuning of other model parameters to ensure the model simulates surface air temperatures with reasonable accuracy for the pre-industrial. For more details see Appendix A1 of the revised manuscript.

[Figure]

*Figure 1: (a) Different values of zonal mean cloud cover including NorESM2 (ZEMBA-PI in grey), CESM2 (ZEMBA-PI$_{CESM2}$ in blue), MRI-ESM2 ( ZEMBA-PI$_{MRI-ESM2}$ in green), CERES 2005-2015 ( ZEMBA-PI$_{CERES}$ in black), ERA5 1940-1970 ( ZEMBA-PI$_{ERA5}$ in red); (b) the zonal mean surface air temperature simulated by ZEMBA in response to these different cloud covers; (c) the differences in zonal mean cloud cover relative to NorESM2 PI cloud cover; and (d) the differences in zonal mean surface air temperature (caused by using different cloud cover fractions) relative to the standard ZEMBA-PI simulation. Pre-industrial cloud cover from CERES and MRI-ESM2 is taken from the Earth System Grid Federation at https://esgf-node.llnl.gov/search/cmip6/ (last accessed: 25/10/2024). ERA5 cloud cover is taken from \cite{ERA5} (last accessed on 25/10/2024) and CERES is taken from https://ceres.larc.nasa.gov/data/ (last accessed: 25/10/2024).*

Regarding the tuning procedure, it involves applying the model parameters chosen in Bintanja (1997), or those modified in the subsequent study by Stap et al. (2014), before adjusting the parameters to ensure a good match for simulations of PI surface air temperature. In a previous iteration of the model, we performed a large ensemble simulation to optimise the values for the key model parameters. However, we did not repeat this procedure for the latest version of the model, although we note the main findings regarding a strong sensitivity to cloud cover parameters are the same as those shown below. As for the chosen diffusion coefficient Da, we acknowledge it has been adjusted in both hemispheres to ensure a better PI simulation. Therefore, any deficiencies in the simplified radiation parameterization, ocean heat transport model or surface albedo, etc., have likely been compensated for by changing Da. In the Numeric subsection of the Methods, we have added this text providing additional details on the tuning procedure [**line 346**]:

*"Values for key model parameters are summarized in Table 1. These are based on values used in previous studies using an EBM which formed the basis of ZEMBA (Bintanja, 1997; Stap et al., 2014), but with small adjustments to improve the simulated pre-industrial zonal mean temperature. The coefficient for atmospheric heat transport (Da) has been modified in both hemispheres to improve the simulated polar temperature. A parameter sensitivity study is included in Appendix A, demonstrating that the cloud cover parameters significantly impact the simulated climate. In particular, ZEMBA is very sensitive to the globally-averaged cloud optical depth parameter (τ), which has been used as a tuning parameter to adjust the radiation budget to match the present-day (Bintanja, 1997; Stap et al., 2014). The full list of all model variables, parameters and constants is included in Appendix B."*

In the revised manuscript we have included additional sensitivity experiments (in the appendix) to assess the relative impact of internal model parameters on the simulated climate. As noted in Bintanja (1997), it is challenging to define a realistic range or uncertainty of certain model parameters which are representative of processes averaged across an entire latitudinal band (e.g. ocean heat transport) or even globally (e.g. turbulent heat flux coefficient, cloud optical depth). Therefore, we decided to replicate the

large perturbations to model parameters made by Bintanja (1997) to study the sensitivity of the model to key parameters and contrast this to the earlier study by Bintanja (1997). In Appendix A1 we include a section detailing the sensitivity of the model to internal model parameters with the following figure, showing a heightened sensitivity to cloud parameters, and especially the globally averaged cloud optical depth parameter. For more details see Appendix A2 of the revised manuscript.

[Figure]

*Figure 2: Anomalies in global mean surface air temperature, global mean ocean temperature and the air temperature difference between the equatorial (0◦-10◦) and polar regions (80◦-90◦) for changes in the solar constant (SO), atmospheric CO2 level (CO2), cloud amount, cloud optical depth (τ), ocean diffusion coefficient for horizontal eddy/gyre heat transport (Do), ocean diffusion coefficient for vertical heat transport (Dz), coefficient for turbulent heat fluxes (κ), sea ice thickness (dsi) and a Hadley cell parameter (λ).*

**Additionally, some clarification on the impact and inclusion of the seasonal cycle would be useful. Is the seasonal cycle being solely driven by insolation changes, or do other parameters change as well? Are simulated climate significantly different if annual-mean insolation values are used? In Siler et al. (2018), only annual-mean precipitation and evaporation patterns were modeled (not seasonal variations); given ZEMBA's underestimation of snow coverage over land (Fig. 5), is the inclusion of seasonal hydrology reasonable?**

The seasonal cycle is driven exclusively by changes in insolation. No other input parameter in the model contains a seasonal cycle. In the radiative fluxes subsection of the Methods, we have clarified this with the sentence [**line 129**]: *"The seasonal cycle is driven exclusively by changes in insolation."*

To respond to the question as to whether the simulated climate is significantly different if ZEMBA is instead forced by annual-mean insolation, we have repeated the PI, LGM and $2xCO_2$ experiments but with annual-mean insolation at every latitude. Figure 3 below shows the differences in insolation (Fig. 1a-d), surface air temperature (Fig. 3e-h) and sea ice (Fig. 3i-l) between ZEMBA forced with a seasonal cycle and ZEMBA forced with annual-mean insolation.

[Figure]

*Figure3: Anomalies in insolation (top row), surface air temperature (middle row) and sea ice (bottom row) for ZEMBA forced with the full seasonal cycle in insolation, relative to the ZEMBA forced with annual mean insolation. Anomalies in the seasonal cycle of each variable are shown for the LGM (a,e,i), PI (b,f,j) and 2xCO2 (c,g,k) simulation of ZEMBA. Additionally, changes in the annual-mean of each variable, for each experiment, is shown on the right-hand-side (d,h,l).*

For each experiment, the absence of the seasonal cycle leads to significant global cooling, ranging from 2.45 and 3.34°C. The higher temperatures induced when the model is forced by seasonal insolation has also been noted by Bintanja (1997), which they explain by the concentration of solar radiation at the high latitudes in the summer months, when both zenith angles and snow-cover is low, thereby reducing the surface albedo and promoting further absorption of shortwave radiation. To account for this, Bintanja (1997) retuned the annual mean version of their EBM with a lower cloud optical depth, to ensure it produces a somewhat similar climate to the seasonal version of the mode. Overall, the inclusion of the seasonal cycle significantly affects the simulated climate and ZEMBA appears particularly sensitive to summer insolation.

To address this finding in the revised manuscript, we have added the following text in the radiative fluxes subsection of the Methods [**line 129**]:

*"The absence of a seasonal insolation cycle results in a markedly colder climate. As noted by Bintanja (1997), employing an annual-mean version of their EBM results in insolation no longer being concentrated in the summer months, when lower zenith angles and reduced snow cover enhances the absorption of shortwave radiation."*

**Line 95 – "its" not "it's"**

Amended in the new manuscript.

**Line 555 – there is a model available that couples a carbon cycle to an EBM with a hydrologic cycle, and simulates ice sheet growth and decay (Kukla et al; "All aboard! Earth system investigations with the CH2O-CHOO TRAIN v1.0"). A reference should be included, and**

**would recommend rephrasing "There are comparatively few EBMs which incorporate a hydrological cycle (Jentsch, 1991) and none – to our knowledge – used for studies of glacial-interglacial cycles."**

We have rephrased this sentence in the revised manuscript as follows [**line 585**]: *"There are comparatively few EBMs which incorporate a hydrological cycle (Jentsch, 1991; Kukla et al., 2023)."*

In the same paragraph, we later reference the Kukla et al., 2023 paper [**line 590**]: *"Kukla et al. (2023) introduced a moist static energy balance model with a hydrological cycle coupled to a carbon cycle model, suited for studying the long-term relationship between the carbon cycle, hydrological cycle and climate. However, the modelled precipitation does not impact the surface albedo, which is instead simplified as a function of surface temperature, and, like Jentsch (1991), it lacks the seasonal insolation cycle necessary to study the climate response to Milankovitch cycles"*

---

## Author Response (AR2)

List of changes made in the manuscript following reviewer comments:

- Lines 118 & 123: Correction to Equation 1 and the text regarding the symbol for the latent heat of fusion.
- Line 129: Additional text with more details on the seasonal cycle of ZEMBA and its impact on the simulated climate
- Line 162: Additional text clarifying the value for water availability (W) over land and ocean for the latent heat fluxes
- Paragraph beginning at Line 181: We clarify that for calculations of snowfall, surface air temperature is corrected for using a prescribed zonal-mean elevation. We also provide more details on $f_{sf}$ as the geographic fractional area of the grid box over which precipitation falls as snow.
- Lines 228-232: Changes to Equations (18) and (19) to make equations for sensible and latent heat fluxes by the atmosphere more explicit.
- Lines 243 & 246 & 252: Changes to symbol for snowpack thickness over land to $d_{sc}$
- Line 265: Additional text clarifying some assumptions regarding the albedo over land and the absence of vegetation influence or feedbacks.
- Line 296: Additional text that "z" represents the vertical coordinate for the ocean domain of the model.
- Line 326: Deletion of repeated 'the' word
- Line 327: Specification for sea ice thickness.
- Paragraph beginning at Line 340: We removed sentence on sea ice thickness which is now stated earlier. We provide more detail on the tuning procedure, and reference additional sensitivity experiments on model parameters presented in the Appendix, which shows a heightened sensitivity to the cloud parameters. We also reference a table in the Appendix of all model parameters referenced in the text.
- Changes to Table 1 to include the prescribed sea ice thickness ($d_{si}$)
- Paragraph beginning at Line 426: We clarify the simulated atmospheric heat transport by NorESM2 is in keeping with estimates based on atmospheric reanalysis.
- Paragraph beginning at Line 535: We discuss more the limitations of ZEMBA regarding the simulated snow cover over land.
- Paragraph beginning at Line 585: We include an additional reference to a study (Kukla et al., 2023) involving an EBM with a hydrological cycle.
- Inclusion of Appendix A beginning at Line 685: We presents some additional sensitivity experiments of ZEMBA. Appendix A1 demonstrates the sensitivity of ZEMBA to the prescribed cloud cover for simulation of the pre-industrial. It includes new Figure A1. Appendix A2 presents the sensitivity of ZEMBA to key model parameters in ZEMBA and includes new Figure A2.
- Inclusion of Appendix B beginning at Line 752: We presents a long table of all model parameter referenced in the manuscript, including their symbols, definitions and units.
- Line 878: New reference to Kukla et al. (2023)
- Line 957: New reference to Trenberth and Fasullo (2017)